# Mapping dopaminergic projections in the human brain with resting-state fMRI

Marianne Oldehinkel[1,2,3]*, Alberto Llera[1,2], Myrthe Faber[1,2,4], Ismael Huertas[5], Jan K Buitelaar[1,2], Bastiaan R Bloem[6], Andre F Marquand[1,2,7], Rick C Helmich[1,6], Koen V Haak[1,2]†, Christian F Beckmann[1,2,8]†

[1]Donders Institute for Brain, Cognition and Behaviour, Radboud University Medical Centre, Nijmegen, Netherlands; [2]Department of Cognitive Neuroscience, Radboud University Medical Centre, Nijmegen, Netherlands; [3]Turner Institute for Brain and Mental Health, School of Psychological Sciences, and Monash Biomedical Imaging, Monash University, Clayton, Australia; [4]Department of Communication and Cognition, Tilburg Centre for Cognition and Communication, Tilburg University, Tilburg, Netherlands; [5]Institute of Biomedicine of Seville (IBiS), Seville, Spain; [6]Department of Neurology, and Centre of Expertise for Parkinson & Movement Disorders, Radboud University Medical Centre, Nijmegen, Netherlands; [7]Centre for Neuroimaging Sciences, Institute of Psychiatry, King's College London, London, United Kingdom; [8]Wellcome Centre for Integrative Neuroimaging (WIN FMRIB), University of Oxford, Oxford, United Kingdom

*For correspondence:
marianne.oldehinkel@donders.
ru.nl

†These authors contributed
equally to this work

Competing interest: See page
20

Reviewing Editor: Shella
Keilholz, Emory University and
Georgia Institute of Technology,
United States

**Abstract** The striatum receives dense dopaminergic projections, making it a key region of the dopaminergic system. Its dysfunction has been implicated in various conditions including Parkinson's disease (PD) and substance use disorder. However, the investigation of dopamine-specific functioning in humans is problematic as current MRI approaches are unable to differentiate between dopaminergic and other projections. Here, we demonstrate that 'connectopic mapping' – a novel approach for characterizing fine-grained, overlapping modes of functional connectivity – can be used to map dopaminergic projections in striatum. We applied connectopic mapping to resting-state functional MRI data of the Human Connectome Project (population cohort; N = 839) and selected the second-order striatal connectivity mode for further analyses. We first validated its specificity to dopaminergic projections by demonstrating a high spatial correlation ($r$ = 0.884) with dopamine transporter availability – a marker of dopaminergic projections – derived from DaT SPECT scans of 209 healthy controls. Next, we obtained the subject-specific second-order modes from 20 controls and 39 PD patients scanned under placebo and under dopamine replacement therapy (L-DOPA), and show that our proposed dopaminergic marker tracks PD diagnosis, symptom severity, and sensitivity to L-DOPA. Finally, across 30 daily alcohol users and 38 daily smokers, we establish strong associations with self-reported alcohol and nicotine use. Our findings provide evidence that the second-order mode of functional connectivity in striatum maps onto dopaminergic projections, tracks inter-individual differences in PD symptom severity and L-DOPA sensitivity, and exhibits strong associations with levels of nicotine and alcohol use, thereby offering a new biomarker for dopamine-related (dys)function in the human brain.

## Editor's evaluation

The authors make a convincing argument that they have found an MRI-based biomarker for dopaminergic input into the striatum. Because the dopaminergic system is involved in neurodegenerative disorders such as Parkinson's disease and also in processing reward signals, the biomarker is likely to

become widely adopted and enable new types of experiments in related fields. In this revision, the authors further demonstrate the specificity of the potential biomarker and its lack of sensitivity to head motion.

## Introduction

The brain's dopamine system plays an important role in a wide range of behavioural and cognitive functions, including movement and reward processing (*Joshua et al., 2009*; *Ruhé et al., 2007*). An integral structure of the dopamine system is the striatum, which receives dense dopaminergic projections from the substantia nigra pars compacta (SNc) and ventral tegmental area (VTA) in the midbrain (*Steiner and Tseng, 2016*). Work in experimental animals has shown that these projections organize along a gradient: dopaminergic neurons in the SNc project preferentially to dorsal caudate and putamen in dorsolateral striatum, while dopaminergic neurons in the VTA project predominantly to the nucleus accumbens (NAcc) in ventromedial striatum (*Steiner and Tseng, 2016*; *Haber, 2014*; *Björklund and Dunnett, 2007*). The projections from the SNc to dorsolateral striatum comprise the nigrostriatal pathway implicated in, for example, the organization of motor planning (*Joshua et al., 2009*; *Faure et al., 2005*). The mesolimbic pathway formed by the projections from the VTA to the NAcc has been associated with reward processing (*Schultz, 2013*; *Wise, 2004*). In accordance with the partial neuroanatomical overlap in striatum, increasing evidence also suggests partial overlap in the function of both pathways (*Haber et al., 2000*; *Everitt and Robbins, 2005*; *Wise, 2009*). Of note, dopaminergic neurons in the VTA not only project to NAcc but also to prefrontal cortex. These cortical projections form the mesocortical pathway associated with reward-related goal-directed behaviours (*Schultz, 2013*; *Wise, 2004*).

In humans, alterations in these dopaminergic projections have been associated with multiple neurological and psychiatric conditions (*DeLong and Wichmann, 2007*; *Money and Stanwood, 2013*). A well-known example is Parkinson's disease (PD), a neurodegenerative disorder characterized by a loss of dopaminergic neurons in the SNc (part of the nigrostriatal pathway; *Fearnley and Lees, 1991*), which frequently causes asymmetric depletion of dopamine in dorsal striatum (first in putamen, later also to a lesser extent in caudate) and leads to impairments in motor as well as a range of nonmotor functions (*Brooks and Piccini, 2006*; *Hornykiewicz, 2008*). Dopaminergic dysfunction has also been implicated in substance use disorder given that addictive substances, such as stimulants, alcohol, and nicotine, increase the release of dopamine in ventral striatum (i.e., mesolimbic pathway; *Laruelle et al., 1995*; *Barrett et al., 2004*; *Nutt et al., 2015*).

Despite the important role of the dopamine system in human brain function and its implication in disease, knowledge about this neurotransmitter system is limited and mainly based on experimental work in animals. The investigation of dopaminergic functioning in vivo in the human brain is challenging, although the nuclear imaging techniques position emission tomography (PET) and single photon emission computed tomography (SPECT) can be used for this purpose (*Blake et al., 2003*; *Volkow et al., 1996*). Imaging of the density of the dopamine transporter (DaT) using SPECT has become a popular tool to assist in the differential diagnosis of PD as loss of dopaminergic neurons in PD is accompanied by a loss in DaT in striatum, as opposed to lookalike conditions such as dystonic tremor where the DaT signal remains intact (*Brooks, 2016*). Tracking the loss of DaT signal over time has also been proposed as a progression biomarker for PD (*Brooks, 2016*). Indeed, DaT reuptakes dopamine from the synaptic cleft after its release and is highly expressed in the terminals of dopaminergic neurons projecting from the midbrain to striatum (*Brooks, 2016*). Therefore, DaT SPECT imaging can be used to image dopaminergic projections in striatum. However, the radiation exposure and costs of PET/SPECT combined with the low spatial resolution of the scan limit widespread implementation in human brain research and in clinical practice.

In this work, we hypothesize that inter-individual differences in DaT availability induce inter-individual variations in the synchronicity of functional activity in the brain, and therefore, that dopaminergic projections in the human striatum can also be mapped using blood-oxygen-level-dependent (BOLD) functional MRI (fMRI) measured at rest. We employ a 'connectopic mapping' data analysis approach to disentangle striatal connectivity into multiple overlapping spatial 'modes' in order to dissect the complex mixture of efferent and afferent connections of the striatum to multiple cortical and subcortical systems (that map onto different neurobiological systems and associated functions;

*Haak et al., 2018*). In previous work, we already showed that the dominant (zeroth-order) mode represents its basic anatomical subdivisions, while the first-order mode maps on to a ventromedial-to-dorsolateral gradient associated with goal-directed behaviour in cortex (*Marquand et al., 2017*) that has been described previously on the basis of tract-tracing work in non-human primates (*Haber and Knutson, 2010*). Here, we demonstrate –by conducting a series of analyses across different data-sets – that the second-order mode of gradual spatial variations in the BOLD connectivity pattern reflects DaT availability in the striatum. We furthermore reveal that this mode tracks inter-individual differences in symptom severity in PD patients, is sensitive to acute dopaminergic modulation (L-DOPA administration), and exhibits strong associations with levels of nicotine and alcohol use in a population-based cohort. Hereby, we provide compelling evidence that this connectivity mode tracks inter-individual differences in dopaminergic projections, and as such, offers a new biomarker for investigating dopamine-related dysfunction across various neurological and psychiatric disorders.

## Results

### Striatal connection topographies map onto DaT availability

For our first analysis, we applied connectopic mapping (*Haak et al., 2018*) to resting-state fMRI data from 839 participants of the Human Connectome Project (HCP; *Van Essen et al., 2013*). Connectopic mapping extracts the dominant modes of functional connectivity change (or connection topographies) within the striatum based on a Laplacian eigenmap decomposition of the similarity matrix derived from functional connectivity (i.e., Pearson correlations) computed between each striatal voxel and the rest of the brain. It provides reproducible and parsimonious representations of overlapping connection topographies at both the group level and at the level of individual subjects. The connectopic mapping approach is detailed in Materials and methods, but a summary of this procedure can be found in *Figure 1*.

For all analyses described in this paper, connectopic mapping was applied to the left and right putamen and caudate-NAcc subregions separately to increase regional specificity and the second-order striatal connectivity mode was selected for each of the four striatal regions of interest (ROIs). A spatial statistical model, that is, a trend surface model (TSM; *Gelfand et al., 2010*), was fitted to both the group-level and the subject-specific connectivity modes to obtain a small set of coefficients summarizing each of the four striatal modes in the X, Y, and Z axes of MNI152 coordinate space, which we used for statistical analyses. A scree test (*Cattell, 1966*) indicated that a quadratic model (i.e., consisting of six TSM coefficients) provided the best fit for the second-order connectivity mode in putamen and a quartic model (12 TSM coefficients) was found to provide the best fit for the second-order connectivity mode in caudate-NAcc region.

The subject-specific second-order striatal connectivity modes were highly consistent across the two fMRI sessions (mean ± SD: $\rho$ = 0.98 ± 0.07; averaged across all four subregions) of the HCP dataset, which is in line with what we have demonstrated previously for other brain regions and for the zeroth-order and first-order mode of connectivity in striatum. Furthermore, interclass correlation (ICC(2,k)), which indexes measurement consistency for a putative biomarker (*Shrout and Fleiss, 1979*; *Koo and Li, 2016*), showed excellent reproducibility of the subject-specific connectivity modes, while still being sensitive to inter-individual differences (see *Table 1*). Both the variations across subjects and the reproducibility within subjects are illustrated in *Figure 2—figure supplement 3*.

The group-level second-order connectivity mode across striatum is displayed in *Figure 2* (second row). The modes for left and right putamen and caudate-NAcc have been combined in this figure (i.e., the four ROIs were loaded in FslView simultaneously from which the below figures were derived) to aid in visualization and for later comparison to the DaT SPECT scan. The second-order connectivity mode comprises a gradient from the dorsal putamen and dorsal caudate (shown in red) to the ventral putamen and ventral caudate including the NAcc (shown in blue). This coding indicates that the dorsal putamen and dorsal caudate exhibit a connectivity pattern with the rest of the brain that is similar to each other but different from the ventral putamen and ventral caudate and vice versa. This striatal connectivity pattern might thus correspond with the gradient of mesolimbic and nigrostriatal dopaminergic projections to striatum (ventral vs. dorsal striatum) well described by track-tracing studies in rodents and non-human primates (*Steiner and Tseng, 2016*; *Haber, 2014*; *Björklund and Dunnett, 2007*). We therefore investigated its spatial correspondence to DaT SPECT-derived DaT

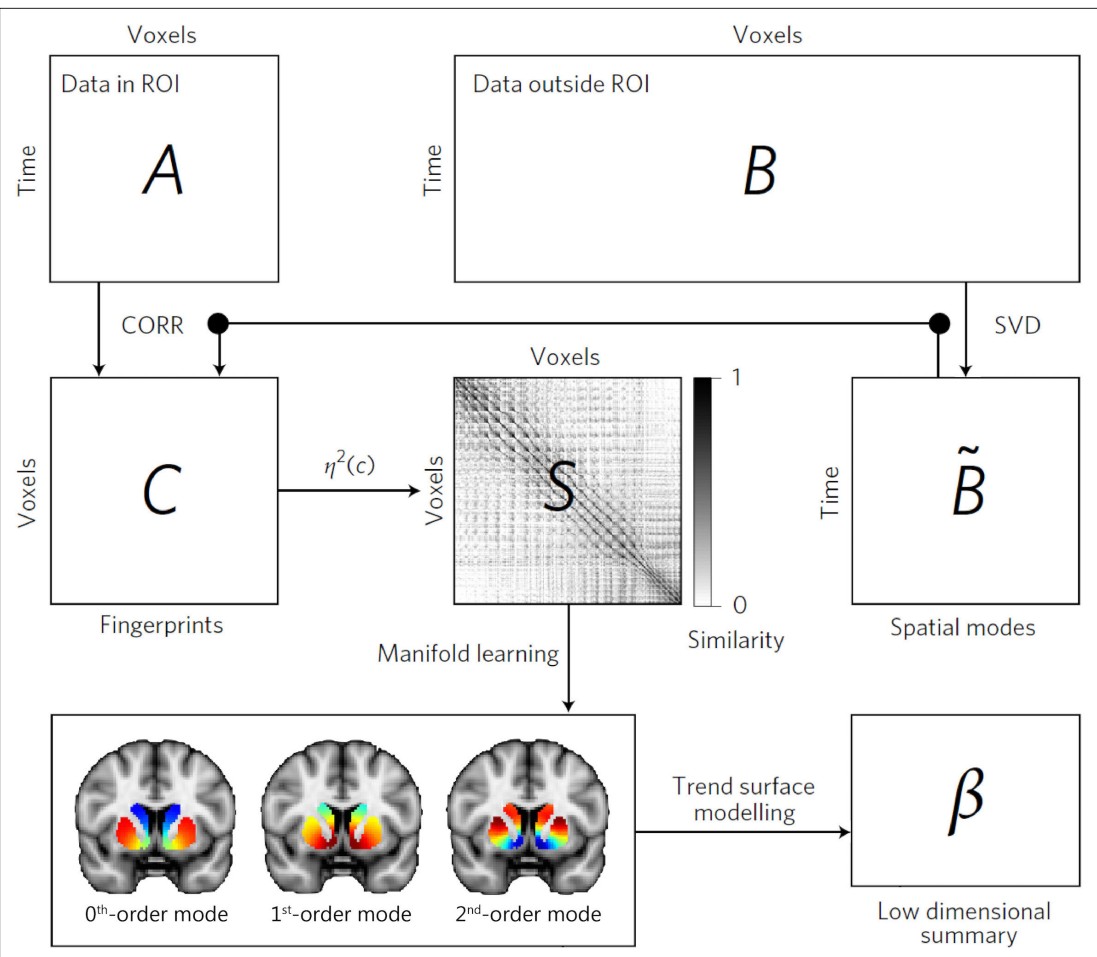

**Figure 1.** The connectopic mapping pipeline. The functional MRI (fMRI) time-series data from a predefined region of interest (ROI), here the striatum, are rearranged into a time-by-voxels matrix *A*, as are the time series from all voxels outside the ROI (matrix *B*). For reasons of computational tractability, the dimensionality of *B* is losslessly reduced using singular value decomposition (SVD), yielding ~*B*. For every voxel within the ROI, its connectivity fingerprint is computed as the Pearson's correlation (CORR) between the voxel-wise time-series and the SVD-transformed data, yielding matrix *C*. Then similarity between voxels is computed using the $\eta^2$ coefficient, resulting in matrix *S*. Manifold learning using Laplacian eigenmaps is then applied to this matrix, yielding a set of overlapping, but independent, connection topographies or 'connectivity modes' that together describe the functional organization of the striatum. These connection topographies indicate how the connectivity profile with the rest of the brain changes across striatum. Voxels that have similar colours in these connectivity modes have similar connectivity patterns with the rest of the brain. Finally, trend surface modelling is applied to summarize the connectivity modes by fitting a set of trend coefficients (*β*) that optimally combine a set of spatial polynomial basis functions. See *Haak et al., 2018* for further details.

availability in striatum, which is assumed to be an index of dopaminergic projections. To this end, we averaged across DaT SPECT images obtained from 209 healthy control participants from the Parkinson's Progression Markers Initiative (PPMI) dataset (*Marek et al., 2011*). As can be observed in *Figure 2*, the group-level second-order striatal connectivity mode indeed displays a remarkably high similarity with the group-level DaT availability in striatum, as quantified by a spatial voxel-wise correlation of $r = 0.884$ (p<0.001), thereby providing the first evidence for an fMRI-derived striatal connectivity marker strongly associated with dopaminergic projections into striatum. A high correlation ($r = 0.925$, p<0.001) is also present between the orthogonal TSM coefficients modelling the group-level second-order connectivity mode and the group-level DaT SPECT scan across the striatum, providing more evidence that the second-order connectivity mode maps onto dopaminergic projections. This finding does not strongly depend on the chosen model order, given that repeating this analysis using model order 3 (i.e., a cubic model with nine TSM coefficients for both the putamen and caudate-NAcc regions) resulted in a similar correlation ($r = 0.90$, p<0.0001).

**Table 1.** Interclass correlation coefficients (ICCs) between the two scanning sessions and the session 1 to session 2 within-subject and between-subject spatial correlations.

| Striatal subregion | ICC [bootstrapped 95% CI] | Within-subject correlation | Between-subject correlation | Within vs. between permutation test ($N_{perm}$ = 10,000) |
|---|---|---|---|---|
| Left putamen | 0.960 [0.951–0.965] | 0.969 | 0.965 | p<0.0001 |
| Right putamen | 0.961 [0.952–0.967] | 0.970 | 0.966 | p<0.0001 |
| Left caudate-NAcc | 0.974 [0.968–0.978] | 0.981 | 0.976 | p<0.0001 |
| Right caudate-NAcc | 0.974 [0.968–0.978] | 0.981 | 0.977 | p<0.0001 |

CI = confidence interval; NAcc = nucleus accumbens.

Finally, to further demonstrate the high specificity of the second-order connectivity mode to the DaT SPECT scan, we computed correlations with the TSM coefficients of all PET scans, tapping into various neurotransmitter systems, included in the publicly available JuSpace toolbox (*Dukart et al., 2021*). *Figure 2—figure supplement 1* reveals that the correlation between the TSM coefficients of the second-order connectivity mode with the DaT SPECT scan is not only highly significant but also significantly higher than the correlations with the TSM coefficients of any other PET scan.

In order to demonstrate that also individual variations in this connectivity mode are associated with individual variations in striatal dopaminergic projections, we further aimed to replicate this mapping at the *within-subject* level in a subsample of PPMI participants (130 datasets from PD patients and 14 from controls) with both DaT SPECT and resting-state fMRI data available. Within a smaller sample of PD patients and controls with good quality connectivity modes (see Appendix 1—Supplementary analyses and *Figure 3—figure supplement 1* for further details), we not only replicated the spatial correspondence between the connectivity mode and DaT SPECT scan at the group level (PD group: *r* = 0.714; control group: *r* = 0.721) but also observed a *within-subject* spatial correlation of 0.58 across the four striatal subregions (0.44> *r* < 0.62; mean = 0.58, 95% CI = [0.56,0.60]) (see *Figure 3*). These findings were not induced by residual head motion (see *Figure 3—figure supplement 2*).

## Striatal connection topographies are altered in PD

The strong association of the second-order striatal connectivity pattern with DaT availability suggests that this resting-state fMRI-derived connectivity mode can be used to assess variability (including disease-related alterations) in dopaminergic projections to the striatum. As such, we hypothesized that the second-order connectivity mode would be altered in PD since this disorder is characterized by progressive degeneration of nigrostriatal dopaminergic neurons. In order to validate this hypothesis, we made use of a separate high-resolution PD dataset (*Dirkx et al., 2019*) including 39 PD patients (19 patients with asymmetric symptoms on the left side of the body, i.e., they were left-dominant; 20 patients were right-dominant) and 20 controls that each underwent two high-resolution resting-state fMRI session (T = 860 ms, 700 time points). Participant characteristics can be found in *Table 2*. During one session, PD patients received dispersible 200/50 mg levodopa-benserazide (L-DOPA), a precursor of dopamine used for the treatment of PD, during the other session they received placebo (dispersible cellulose). Controls did not receive L-DOPA and placebo but just underwent two typical resting-state fMRI sessions under the same scanning protocol. Thus, this dataset did not only allow us to investigate the effects of clinical diagnosis, but also the effects of acute dopaminergic modulation on the underlying striatal connectivity mode. The associations with diagnosis and L-DOPA were investigated in both groups separately (left-dominant, right-dominant PD) because the side of predominant nigrostriatal dopamine depletion likely influences the pattern of striatal connectivity. As before, the second-order striatal connectivity mode was modelled separately for the putamen and caudate-NAcc subregions to increase regional specificity as PD is known to affect the putamen region of the striatum before the caudate-NAcc region (*Kish et al., 1988*). Group differences in the TSM coefficients modelling the putamen and caudate-NAcc subregions were subsequently assessed by conducting an omnibus test of all the TSM coefficients, that is, a likelihood ratio test in the context of a logistic

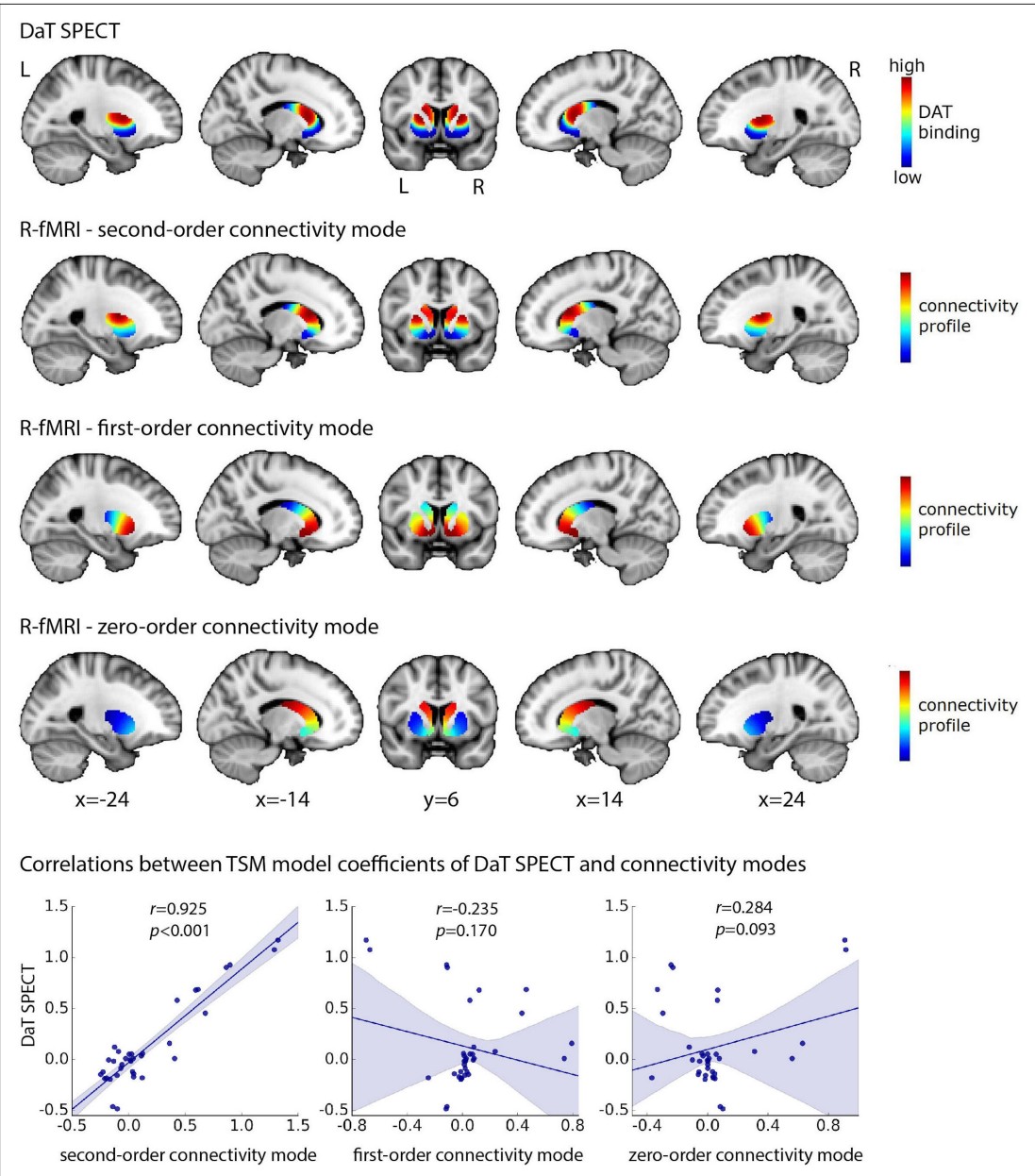

**Figure 2.** High spatial correspondence between the second-order mode of connectivity in striatum and the DaT SPECT image. The figure displays the DaT SPECT image averaged across 209 Parkinson's Progression Markers Initiative (PPMI) controls and the group-level connectivity modes obtained in 839 Human Connectome Project (HCP) subjects. The group-level modes were modelled separately for the left and right putamen and caudate-nucleus accumbens (caudate-NAcc) subregions and have been combined in this figure to aid in visualization. The voxel-wise spatial correlation between the second-order mode of connectivity in striatum and the DaT SPECT image is very high: $r = 0.884$ ($p<0.001$). Similarly, the correlation between the orthogonal trend surface model (TSM) coefficients modelling the second-order connectivity mode and the DaT SPECT scan in striatum is very high: $r = 0.925$ ($p<0.001$, bottom row). R-fMRI, resting-state fMRI; L, left; R, right.

The online version of this article includes the following figure supplement(s) for figure 2:

**Figure supplement 1.** The correlation between trend surface model (TSM) coefficients modelling the second-order connectivity mode and the DaT SPECT scan is highly significant and substantially higher than all other position emission tomography (PET)-derived markers indexing other neurotransmitter systems.

**Figure supplement 2.** The second-order connectivity mode obtained in the 10% lowest and 10% highest movers of the Human Connectome Project (HCP) dataset is comparable to the mode obtained in the full sample.

**Figure supplement 3.** Inter-subject and inter-session (within-subject) variability in the second-order mode of connectivity in striatum.

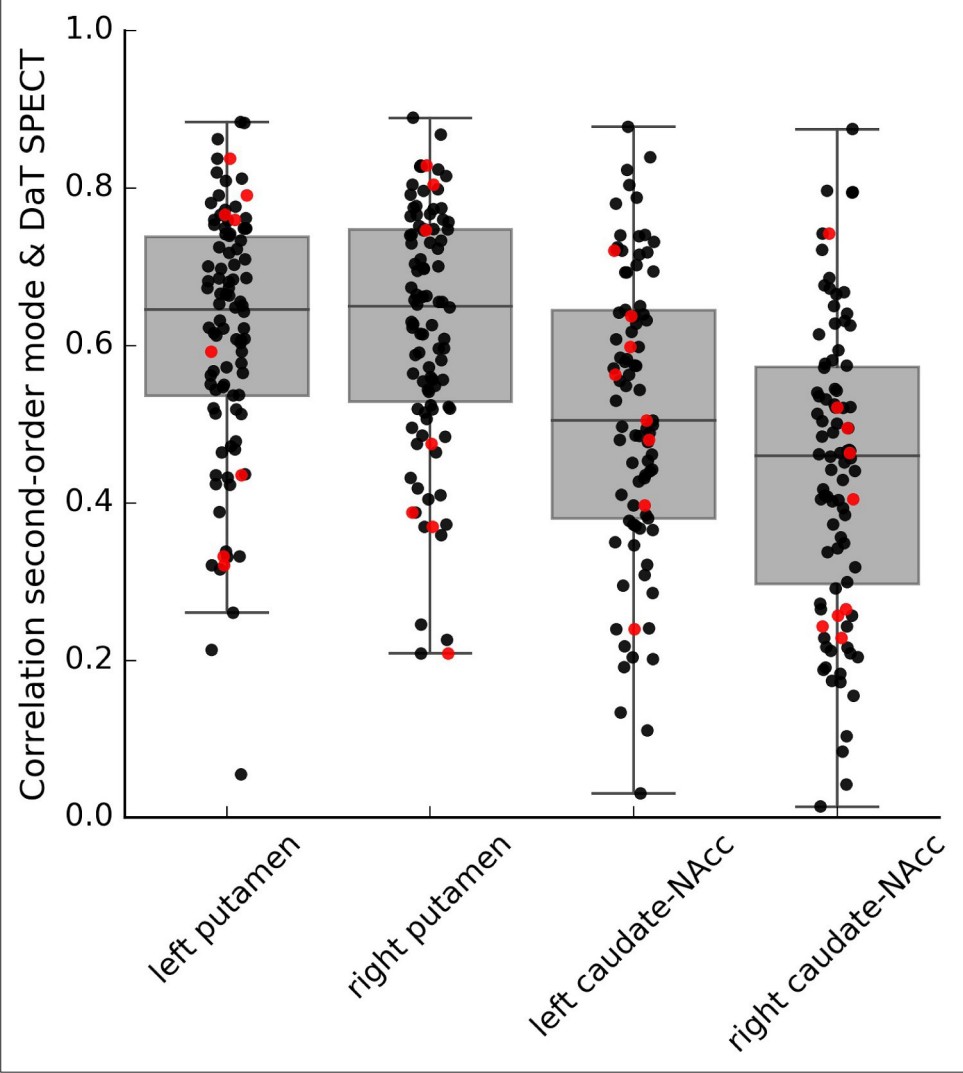

**Figure 3.** Within-subject correlations between the second-order connectivity mode and the DaT SPECT scan from subjects in the Parkinson's Progression Markers Initiative (PPMI) cohort where both resting-state functional MRI (fMRI) and DaT SPECT data is available. These correlations were obtained in a subsample of the PPMI dataset (6–8 datasets from controls and 73–82 datasets from Parkinson's disease patients depending on the striatal subregion) with good-quality connectivity modes as defined by a high spatial correlation ($r > 0.5$) with the group-average Human Connectome Project (HCP) connectivity mode. Red dots represent control participants; black dots represent Parkinson's disease patients.

The online version of this article includes the following figure supplement(s) for figure 3:

**Figure supplement 1.** Spatial correlations of the subject-specific second-order connectivity modes with the mean Human Connectome Project (HCP) connectivity mode and DaT SPECT scan.

**Figure supplement 2.** Within-subject correlations between the second-order connectivity mode and the DaT SPECT scan in a low motion and high motion subsample.

regression. We applied correction for multiple comparisons (two groups: left- and right-dominant PD × 2 striatal subregions: putamen and caudate-NAcc) using a Bonferroni-corrected $\alpha$-level of 0.05/4 = 0.0125. In addition, we also investigated associations between the TSM coefficients and symptom severity across PD patients. To this end, we fitted general linear models (GLMs) that included the TSM coefficients modelling the gradient during the placebo session to predict the total score on the motor section (part III) of the Movement Disorder Society Unified Parkinson's Disease Rating Scale (UPDRS; *Goetz et al., 2008*). This analysis was again conducted separately for the left- and right-dominant

**Table 2.** Participant characteristics.

| Demographic information (mean, SD) | Controls N = 20 | | PD N = 39 | | Test statistic | |
|---|---|---|---|---|---|---|
| Age, years | 61.9 | 10.4 | 60.9 | 10.7 | $t(57) = 0.337$ | NS |
| Sex, male (number, %) | 11 | 55.0% | 16 | 41.0% | $X^2(1) = 0.086$ | NS |
| FAB, total score | 17.6 | 0.67 | 17.3 | 0.97 | $t(57) = 0.23$ | NS |
| Disease duration, years | NA | | 3.96 | 4.57 | | NA |
| L-DOPA equivalent at home (mg/day) | NA | | 467.8 | 227.3 (range: 0–1100) | | NA |
| UPDRS total score (mean, SD) | | | | | | |
| Placebo session | NA | | 40.5 | 16.4 | | |
| L-DOPA session | NA | | 31.9 | 13.1 | $t(38) = 5.58$ | p<0.001 |

For the FAB, lower scores indicate worse functioning; for the UPDRS, higher scores indicate worse functioning. The FAB score was evaluated off medication.

FAB = frontal assessment battery (score 0–18); UPDRS = Unified Parkinson's Disease Rating Scale part III (score 0–132); NS = not significant; NA = not applicable; PD = Parkinson's disease; L-DOPA = levodopa-benserazide.

The online version of this article includes the following source data for table 2:

**Source data 1.** Source data for participant characterists listed in *Table 2*.

groups and separately for putamen and caudate-NAcc subregions, and a Bonferroni-corrected $\alpha$-level of 0.0125 was used for establishing statistical significance.

The PD (placebo session) vs. control group analysis (first session) of the TSM coefficients modelling the second-order striatal connectivity mode revealed a significant difference between the right-dominant PD group and the control group in bilateral putamen (but not in the caudate-NAcc region) of the striatum (see *Figure 4A*; omnibus test of all TSM coefficients: $X^2 = 27.17$, p=0.007). No significant differences were observed between the left-dominant PD group and the control group. Moreover, within the right-dominant patient group, we observed a trend-level association between UPDRS symptom severity scores and the TSM coefficients modelling the putamen under placebo (GLM omnibus test of the TSM coefficients: $X^2 = 22.28$, p=0.035). Post-hoc Pearson correlations revealed that this effect was driven by the quadratic TSM coefficients modelling the striatal connectivity mode in the right putamen in the Y (i.e., anterior-posterior) direction (right $Y^2$: $r = 0.476$, p=0.034) and Z (i.e., superior-inferior) direction (right $Z^2$: $r = -0.460$, p=0.041). This association can be observed in *Figure 4B*, which shows an increase in blue-coded voxels in the second-order striatal connectivity mode as symptom severity increases in PD. This pattern maps very well on the observed decrease in dopaminergic projections as PD becomes clinically more severe, as reflected by higher UPDRS scores. That is, given the spatial similarity of the second-order striatal connectivity mode with the DaT SPECT scan, we can interpret the observed alteration in the connection topography as a decrease in dopaminergic projections to striatum. In *Figure 4B*, this is evident as an increase in blue-coded voxels and a decrease in red-coded voxels as a function of UPDRS symptom severity. Supplementary analyses showed that the observed group differences and associations with symptom severity were independent of age and sex (*Appendix 2—table 1*).

## Striatal connection topographies are sensitive to acute dopaminergic modulation

After demonstrating that the second-order connectivity mode in putamen is indeed altered in PD, we investigated whether it was also sensitive to the acute effects of the dopamine precursor L-DOPA. We assessed differences in this striatal connectivity mode between the placebo and L-DOPA session in both PD groups by conducting an omnibus test of all the TSM coefficients, that is, a likelihood ratio test in the context of a logistic regression. We applied correction for multiple comparisons (two groups: left- and right-dominant PD × 2 striatal subregions: putamen and caudate-NAcc) using a Bonferroni-corrected $\alpha$-level of 0.05/4 = 0.0125. These tests did not reveal significant differences between the placebo and L-DOPA session in the putamen or the caudate-NAcc region. However,

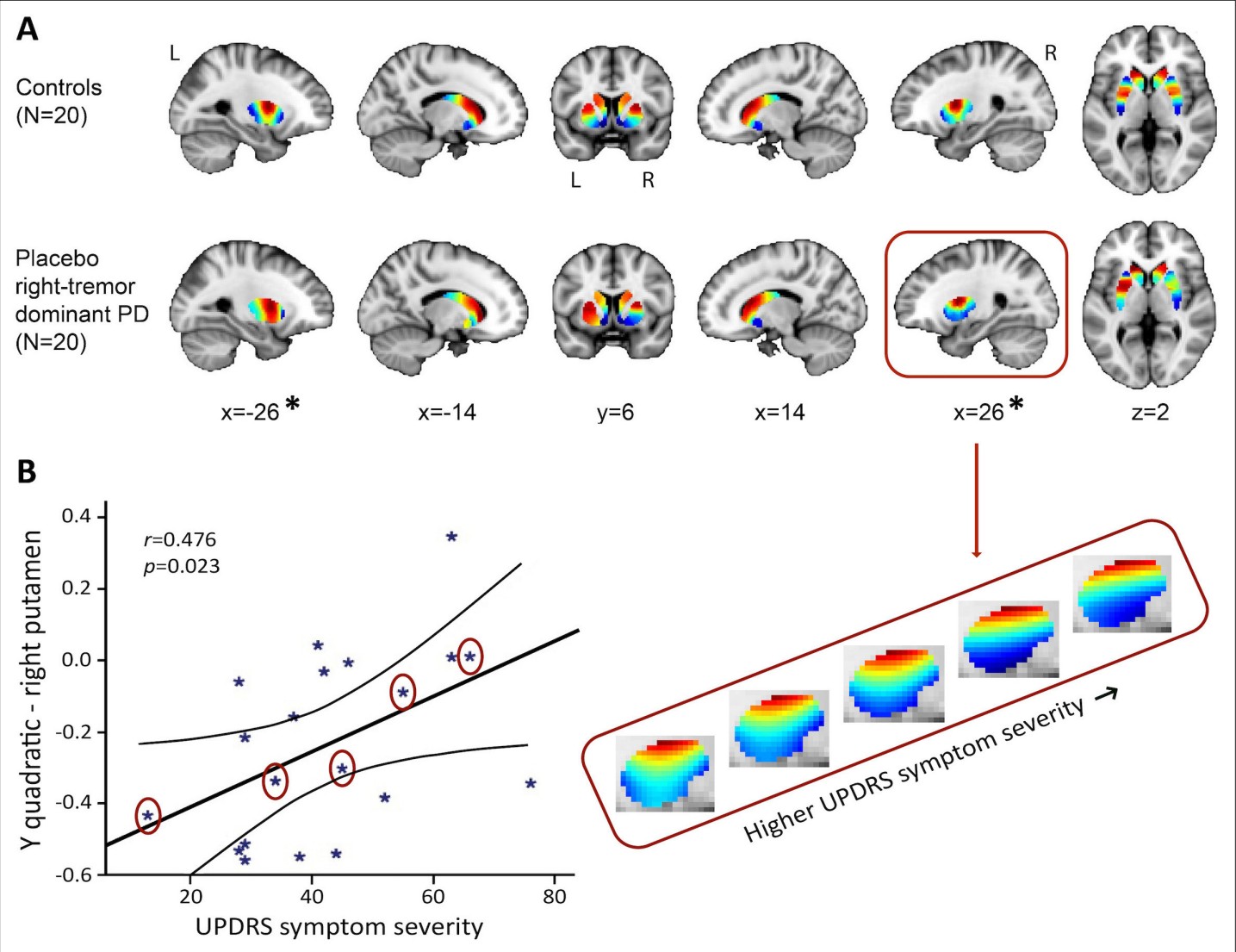

**Figure 4.** The second-order striatal connectivity mode is altered in right-dominant Parkinson's disease. (**A**) Significant difference between the control group and the right-dominant Parkinson's disease group under placebo in putamen (omnibus test of all trend surface model [TSM] coefficients for putamen: $X^2 = 27.17$, p=0.007). Images represent the mean connectivity modes across each of the investigated groups. The slices at MNI coordinates x = –26 and x = 26, respectively, show views of the striatal connectivity mode across left and right putamen where the connectivity mode is significantly different between groups (*); the slices at MNI coordinates x = –14 and x = 14, respectively, show views of the mode across left and right caudate-nucleus accumbens (caudate-NAcc) (no significant difference). (**B**) Trend-level association between the Unified Parkinson's Disease Rating Scale (UPDRS) symptom severity score and the TSM coefficients modelling the second-order connectivity mode in putamen under placebo across patients in the right-dominant Parkinson's disease group (general linear model [GLM] omnibus test of all TSM coefficients: $X^2 = 22.28$, p=0.035). Post-hoc Pearson correlations revealed that this effect was driven by the quadratic TSM coefficients modelling the striatal connectivity mode in the right putamen in the Y (i.e., anterior-posterior) direction (right $Y^2$: r = 0.476, p=0.034) and Z (i.e., superior-inferior) direction (right $Z^2$: r = −0.460, p=0.041). The correlation between UPDRS symptom severity scores is displayed for the right $Y^2$ coefficient. To visualize this association, the reconstructed second-order connectivity mode in the right putamen is shown for five Parkinson's disease patients (data points circled) with increasing UPDRS symptom severity scores.

treatment response to L-DOPA is known to differ among PD patients. To take this variability across patients into account, we conducted GLM analyses relating differences in the L-DOPA-induced change (difference between L-DOPA and placebo session) in the second-order striatal connectivity mode to differences in treatment response. These analyses showed that in both patient groups significant associations were present between the L-DOPA-induced change in UPDRS symptom severity scores and the L-DOPA-induced change in TSM coefficients in putamen (GLM omnibus test of all TSM coefficients in the right-dominant patient group: $X^2 = 25.48$, p=0.012; in the left-dominant patient group:

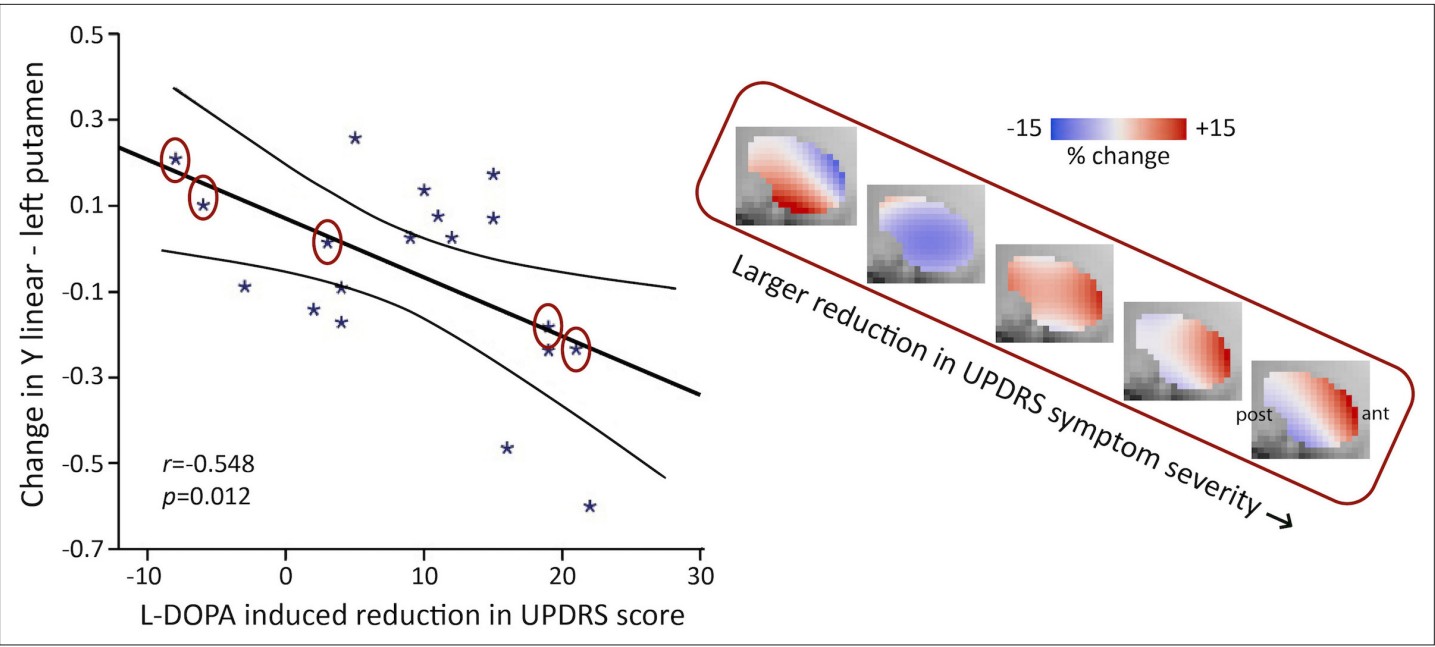

**Figure 5.** Levodopa-benserazide (L-DOPA)-induced reduction in Unified Parkinson's Disease Rating Scale (UPDRS) symptom severity score is associated with the L-DOPA-induced change in the second-order mode of connectivity in putamen in right-dominant Parkinson's disease (PD) (general linear model [GLM] omnibus test of all trend surface model [TSM] coefficients modelling putamen: $X^2$ = 25.48, p=0.012). Post-hoc Pearson correlations revealed that this effect was driven by the linear TSM coefficient modelling the striatal connectivity mode in the left putamen in the Y direction (left $Y^1$: r = −0.548, p=0.012). To visualize this association, the difference in the reconstructed second-order connectivity modes between the placebo and L-DOPA session is shown for the left putamen (at slice X = −24) for five PD patients (data points circled). Red-coded voxels are hypothesized to map onto an increase in dopaminergic connectivity, blue-coded voxels onto a decrease. A significant effect was also observed for the left-dominant PD group (GLM omnibus test of all TSM coefficients: $X^2$ = 34.07, p=0.001), but as post-hoc Pearson correlations did not reveal significant associations with one of the individual TSM coefficients in this group, this association is not shown. ant, anterior; post, posterior putamen.

$X^2$ = 34.07, p=0.001). Post-hoc Pearson correlations revealed that these effects were driven by the linear TSM coefficient ($Y^1$) modelling the striatal connectivity mode in putamen in the Y (i.e., anterior-posterior) direction (right-dominant PD: left $Y^1$: r = −0.548, p=0.012; left-dominant PD: right $Y^1$: r = 0.345, p=0.15). As can be seen in **Figure 5**, a larger L-DOPA-induced reduction in UPDRS scores is associated with a larger positive change (i.e., an increase in red-coded voxels) in the superior-anterior part of the putamen, which we hypothesize maps onto an increase in dopamine-related connectivity. Supplementary analyses demonstrated that the observed effects of L-DOPA were independent of age and sex (see **Appendix 2—table 1**).

## Striatal connection topographies are associated with the amount of substance use

Finally, dopaminergic signalling is also implicated in reward processing, alterations of which have been associated with substance use and compulsive behaviours (**Laruelle et al., 1995**; **Barrett et al., 2004**; **Nutt et al., 2015**). As such, we investigated whether the second-order striatal connectivity mode was also associated with tobacco and alcohol use. These quantities are amongst the set of demographic variates available from the HCP. In order to increase specificity and also in order to transcend analysis from a categorical comparison to a continuous characterization predicting the relative amount of substance usage, we consider here a subset of high tobacco and alcohol users. Specifically, we selected otherwise drug-naïve HCP participants who reported to have consumed ≥3 light and/or ≥ 1 heavy alcoholic units per day during the week preceding the scan (N = 30) and participants reporting to have smoked ≥5 cigarettes every day during the week preceding the scan (N = 38). GLM analyses investigating associations between the TSM coefficients modelling the second-order striatal connectivity mode and the amount of use over the past 7 days were conducted separately for the alcohol users and smokers and separately for putamen and caudate-NAcc subregions. We applied correction

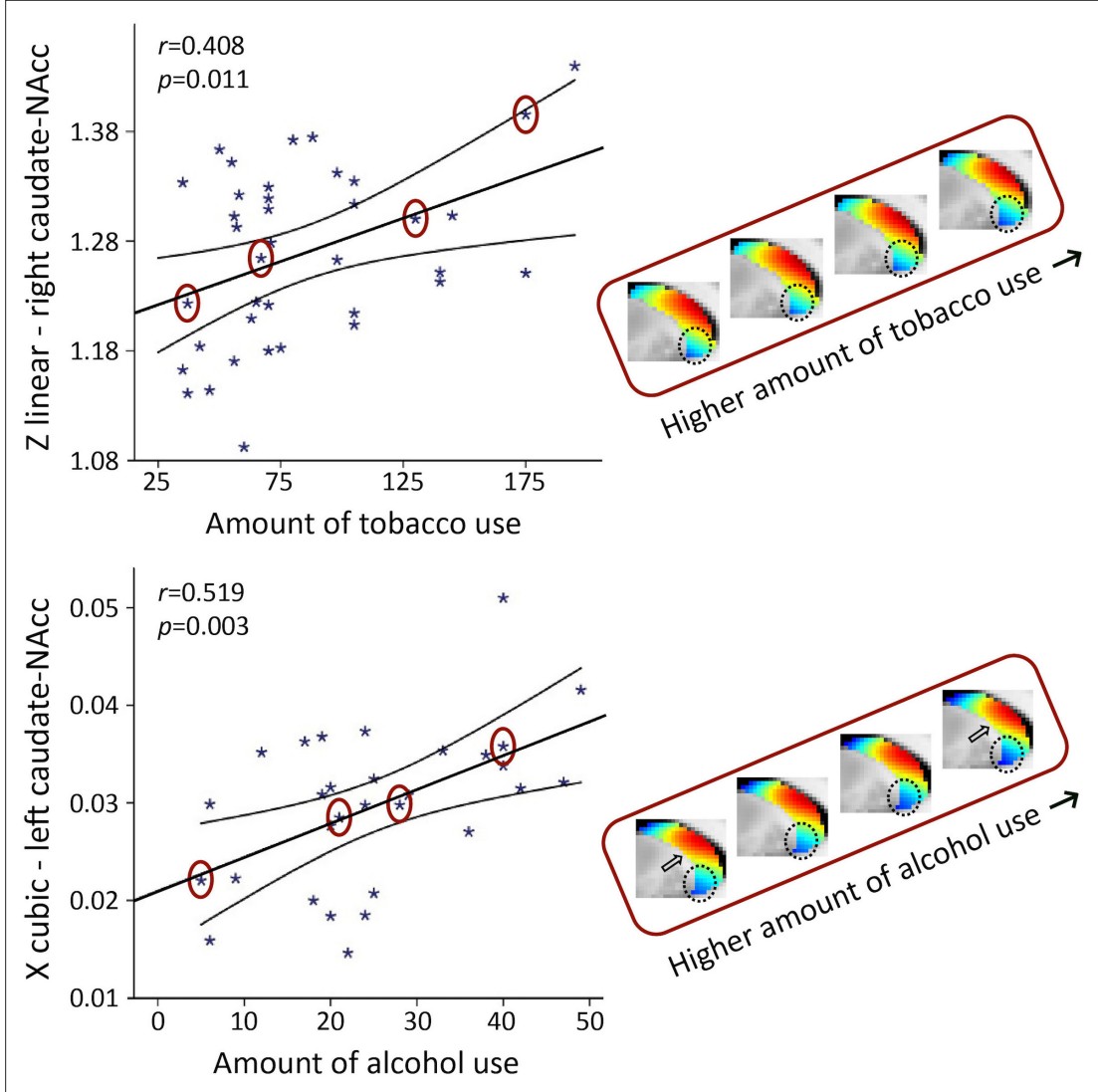

**Figure 6.** The second-order mode of connectivity in striatum is associated with the amount of tobacco use (top) and alcohol use (bottom). Strong associations were observed between the trend surface model (TSM) coefficients modelling the connectivity mode in the caudate-nucleus accumbens (caudate-NAcc) region and the total amount of tobacco use as well as alcohol use over the past week (general linear model [GLM] omnibus test tobacco use: $X^2$ = 49.55, p=0.002; alcohol use: $X^2$ = 64.45, p<0.001). To visualize these relationships, Pearson correlations between one of the significant TSM coefficients and the amount of use are shown as well as the reconstructed second-order connectivity mode in the right caudate-NAcc (at slice X = 14) for four tobacco users and in the left caudate-NAcc (at slice X = −14) for four alcohol users with increasing of amounts of use (data points circled). Circles and arrows indicate where in the connectivity mode tobacco and alcohol use-related changes can be observed. Correlation plots for the other TSM coefficients can be found in *Figure 6—figure supplements 1 and 2*.

The online version of this article includes the following figure supplement(s) for figure 6:

**Figure supplement 1.** The second-order mode of connectivity in striatum is associated with the amount of tobacco use.

**Figure supplement 2.** The second-order mode of connectivity in striatum is associated with the amount of alcohol use.

for multiple comparisons using a Bonferroni-corrected $\alpha$-level of 0.0125 (see Materials and methods for more details).

In smokers, we observed a significant association with the total number of cigarettes smoked over the past week for the TSM coefficients modelling the second-order connectivity mode in the caudate-NAcc region ($X^2$ = 49.55, p=0.002), but not for the putamen region. Subsequent computation of the Pearson correlations between the individual TSM coefficients and the amount of use revealed that this association was driven by multiple TSM coefficients in both the left and right caudate-NAcc (left $X^3$: r = −0.409, p=0.011; right $Z^1$: r = 0.408, p=0.011; right $Y^3$: r = 0.367, p=0.024; right $Z^3$: r = −0.451,

p=0.005; right $Y^4$: $r$ = 0.362, p=0.026; see *Figure 6—figure supplement 1*). As can be observed in *Figure 6* (top panel), alterations in the second-order striatal connectivity mode are subtle but consist of an increase in inferior blue-coded voxels in the caudate-NAcc as tobacco use in this population cohort increases.

In the 'heavy' drinkers, we also observed a strong association with the total number of alcoholic drinks consumed over the past week for the TSM coefficients modelling the connectivity mode in the caudate-NAcc region ($X^2$ = 64.45, p<0.001), but again not for the putamen region. Subsequent computation of the Pearson correlations between the individual TSM coefficients and the amount of use revealed that this association was driven by multiple TSM coefficients in both the left and right caudate-NAcc (left $X^1$: $r$ = −0.378, p=0.039; left $Y^1$: $r$ = 0.399, p=0.029; left $Y^2$: $r$ = 0.488, p=0.006; left $Z^2$: $r$ = −0.386, p=0.035; left $X^3$: $r$ = 0.519, p=0.003; left $Y^4$: $r$ = −0.417, p=0.022; right $Z^2$: $r$ = −0.418, p=0.022; right $Z^4$: $r$ = 0.488, p=0.006; see *Figure 6—figure supplement 2*). Similar to tobacco use, *Figure 6* (bottom panel) shows that higher levels of alcohol use are accompanied by a subtle increase in blue-coded voxels as well as a decrease in red-coded voxels in the caudate-NAcc region of the second-order connectivity mode. We argue that these subtle increases in blue-coded voxels (and decrease in red-coded voxels) in high nicotine and alcohol users map onto decreases in dopaminergic connectivity, which corresponds with reported reductions in dopamine release in striatum in patients with nicotine and alcohol dependence (*Nutt et al., 2015*; *Balfour, 2015*). Supplementary analyses revealed that the associations with the amount of tobacco and alcohol use persisted under different usage thresholds and were independent of age and sex (see *Appendix 2—tables 1–3*). Finally, five subjects were included in both the tobacco and alcohol use analyses, but the associations with tobacco use ($X^2$ = 39.40, p=0.025) and alcohol use ($X^2$ = 62.01, p<0.001) also remained significant after excluding these subjects.

## Discussion

In this work, we provide evidence for a resting-state fMRI-derived biomarker of dopamine function in the human striatum. Specifically, we demonstrated that one particular mode of functional connectivity in the striatum showed a high spatial correspondence to DaT availability, a marker of dopaminergic projections derived from DaT SPECT imaging. This observation generated multiple hypotheses that we validated using both data from PD patients and healthy controls. We showed that this second-order striatal connectivity mode is associated with symptom severity and sensitive to acute dopaminergic modulation by L-DOPA in persons with PD, a disorder characterized by a degeneration of dopaminergic neurons projecting to striatum (*Fearnley and Lees, 1991*; *Brooks and Piccini, 2006*; *Hornykiewicz, 2008*). We also demonstrated that this mode is associated with the amount of tobacco and alcohol use, both of which have been related to alterations in dopaminergic signalling (*Laruelle et al., 1995*; *Barrett et al., 2004*; *Nutt et al., 2015*). As such, our results provide evidence that the second-order mode of functional connectivity in striatum maps onto dopaminergic projections and can be used as a non-invasive biomarker for investigating dopaminergic (dys)function in PD and substance use. While our results still need to be replicated out of sample to warrant immediate application in clinical practice, formal quantification of test–retest reliability already suggests that this gradient approach has very high measurement consistency and therefore lends itself for further investigation into the clinical utility across the various neurological and psychiatric disorders associated with dopaminergic functioning.

By applying connectopic mapping, we shift away from the vast majority of resting-state fMRI studies that employ hard parcellations to investigate functional brain connectivity. Gradient-based approaches such as connectopic mapping were only developed recently, but have already been successfully employed in several studies to investigate functional connectivity in cortical (*Haak et al., 2018*; *Margulies et al., 2016*; *Saadon-Grosman et al., 2020*) and subcortical regions (*Marquand et al., 2017*; *Yang et al., 2020*; *Tian et al., 2020*). Recent work (*Hong et al., 2020*) has furthermore demonstrated that connectivity gradients can generally be obtained with high reproducibility and reliability, and can predict phenotypic variations with higher accuracy than connectivity measures derived from traditional parcellation-based approaches, making them of interest for potential biomarker development. Indeed, hard parcellations only allow investigating the average functional connectivity signal in one or more regions of interest and thereby ignore both the topographic organization of and functional multiplicity in the brain. In contrast, gradient-based approaches do not only enable

characterization of smooth, gradual changes in functional connectivity, but also enable the detection of multiple, overlapping modes of functional connectivity in a region that might exist at the same time (*Haak and Beckmann, 2020*). The work presented here signifies the importance of both features by not only showing that the second-order striatal connectivity mode comprises a smooth gradient from the dorsal putamen and dorsal caudate to the ventral putamen and ventral caudate including the NAcc, but also that this second-order mode – but not the zeroth-order or first-order mode – maps onto DaT availability. In doing so, we are the first to demonstrate a direct mapping between a functional connectivity-derived marker and dopaminergic projections.

DaT is highly expressed in the terminals of dopaminergic neurons projecting from the midbrain to striatum (*Brooks, 2016*). The high spatial correlation ($r = 0.884$) between the group-average second-order connectivity mode in the HCP dataset and the group-average DaT SPECT image in the PPMI dataset as well as the significant within-subject spatial correlation between the connectivity mode and DaT SPECT scan ($r = 0.58$) in PPMI subjects therefore suggests that this connectivity mode maps onto these dopaminergic projections to striatum. We further demonstrated that the association of the second-order connectivity mode with the DaT SPECT scan was stronger than that of all the other investigated PET markers indexing various neurotransmitter systems (see *Figure 2—figure supplement 1*). This figure also shows that correlations of the second-order connectivity mode with dopamine receptors D1 and D2 in striatum, which are present on postsynaptic dopaminergic neurons, were substantially lower and not significant ($r = -0.290$, p=0.086 and $r = 0.241$, p=0.156), suggesting that the second-order connectivity mode is specific to *presynaptic* dopaminergic projections. Animal work has furthermore shown that dopaminergic projections form a gradient with nigrostriatal neurons from SNc projecting predominantly to the dorsolateral striatum (putamen and caudate) and mesolimbic neurons from the VTA projecting predominantly to the ventromedial striatum (NAcc; *Steiner and Tseng, 2016*; *Haber, 2014*; *Björklund and Dunnett, 2007*) representing a functional connectivity gradient formed by the SNc projections to the dorsolateral (putamen/caudate) and VTA projections to the ventromedial striatum (NAcc). Studies demonstrating that the average striatal DaT binding as obtained by DaT SPECT or PET imaging is highly correlated with averaged post-mortem SN cell counts in humans are in support of this view (*Snow et al., 1993*; *Colloby et al., 2012*; *Kraemmer et al., 2014*). However, to our knowledge the relationship between DaT SPECT/PET and VTA cell counts in humans has not been investigated, and future work will thus be necessary to determine the exact relationship between this striatal connectivity mode, DaT availability assessed by DaT SPECT, and dopaminergic projections.

While we were able to replicate the spatial correlation between the second-order connectivity mode and the DaT SPECT scan at the *within-subject* level, this spatial correlation ($r = 0.58$) is not as high as the spatial correlations observed at the group level (i.e., $r = 0.721$ and $r = 0.714$ for PPMI controls and PD patients respectively, and $r = 0.884$ between the DaT SPECT scan in PPMI controls and the connectivity mode in HCP participants). This is not surprising given the relatively low temporal resolution of the resting-state fMRI scan of the PPMI dataset (TR = 2400 ms, 260 volumes). While this resolution is sufficient for typical resting-state fMRI analyses at the group level, the precise delineation of the very fine-grained and overlapping connectivity modes using connectopic mapping at the single-subject level calls for high spatial and temporal resolution data (*Haak et al., 2018*). However, to our knowledge, there is currently no dataset (publicly) available that includes both a high-resolution resting-state fMRI scan and a DaT SPECT scan from the same participants. With respect to *Figure 3*, we further note the difference in the *within-subject* correlation for the putamen ($r = 0.61/0.62$) compared to caudate-NAcc region ($r = 0.51/0.44$). We tentatively speculate that this difference might relate to a stronger and more stable dopamine-related resting-state fMRI signal in putamen compared to caudate-NAcc resulting from more dopaminergic projections to putamen (*Hörtnagl et al., 2020*), and the putamen being larger in size and spatially further away from the ventricles and therefore less susceptible to motion-related artefact than the caudate-NAcc region.

Nevertheless, adding to its association with dopaminergic projections are the alterations of this connectivity mode observed in PD. This disorder is characterized by a loss of nigrostriatal dopaminergic neurons projecting from SNc to the striatum, which is most prominent in the putamen (*Fearnley and Lees, 1991*). Corresponding to the pathology of the condition, we observed a significant difference of this connectivity mode between right-dominant PD patients and control participants in bilateral putamen. Moreover, within this patient group the second-order striatal connectivity mode was

also sensitive to inter-subject variability as revealed by the association with symptom severity. This association is visualized in *Figure 4B*, which shows that portions of the gradient in right putamen that map on low DaT availability (blue) increase as symptom severity in PD increases. We argue that the second-order striatal connectivity mode hereby follows the expected pattern of a reduction in dopaminergic projections to the putamen (as indexed by decreased DaT availability) as symptom severity increases in PD.

While group differences in the connectivity mode were thus present in bilateral putamen, the association with symptom severity was driven by the TSM coefficients modelling the right putamen. This latter finding might appear counterintuitive as a tremor dominant to the right side of the body in PD (i.e., right-dominant PD) is assumed to correspond with a dopamine depletion that is dominant to the contralateral striatum, that is, left striatum. However, post-mortem studies have reported so-called flooring effects by demonstrating a complete absence of dopaminergic fibres in the dorsal putamen in the most affected hemisphere in PD patients ≥ 4 years after disease onset (*Kordower et al., 2013*). Furthermore, resting-state fMRI studies have reported larger PD vs. control group differences in the anterior putamen of the lesser affected compared to the more affected hemisphere (*Helmich et al., 2010*). Taken together, these findings might suggest that in many patients with right-dominant PD in our sample (mean disease duration is 3.96 years) dopamine-related connectivity in the contralateral left putamen is showing flooring effects, making associations with symptoms only detectable in the right putamen. Future research is necessary to confirm this hypothesis. It should also be noted that we did not find a significant difference between controls and the left-dominant PD group. Since there is no evidence that different mechanisms underlie left-dominant and right-dominant PD (apart from the difference in the most-affected hemisphere), this might be a power issue that requires further investigation.

Not only was the second-order mode of connectivity in striatum sensitive to variability in symptom severity in a clinical cohort, but also to behavioural variability associated with self-reported alcohol and tobacco use in a healthy, non-clinical population. Substance use has frequently been associated with alterations in dopamine release in the ventromedial striatum (NAcc), part of the mesolimbic dopaminergic pathway (*Laruelle et al., 1995*; *Barrett et al., 2004*; *Nutt et al., 2015*). Corresponding with these findings, we observed significant associations with the amount of alcohol and tobacco use over the past week in the caudate-NAcc region, but not in the putamen region of the second-order striatal connectivity mode. More specifically, we observed subtle increases in blue-coded voxels and decreases in red-coded voxels as substance use increased, suggesting decreased DaT availability or more generally decreased dopaminergic signalling in the caudate-NAcc region in high nicotine and alcohol users. These results are consistent with findings from previous DaT SPECT and PET studies reporting reductions in striatal DaT availability in patients with alcohol dependence (*Grover et al., 2020*; *Yen et al., 2016*; *Laine et al., 1999*; *Repo et al., 1999*) and nicotine dependence (*Yang et al., 2008*). However, a limitation that should be mentioned is that the resting-state fMRI sequence of the HCP dataset has not optimized for subcortical brain regions.

As such, not only are these alterations in PD and high alcohol and tobacco users of the HCP dataset consistent with the hypothesis that the second-order striatal connectivity mode reflects dopaminergic projections, the alterations are also specific to the hypothesized striatal subregions and dopaminergic pathways. That is, we found that PD –a disorder characterized by death of nigrostriatal dopaminergic neurons leading to motor impairments– was associated with connectivity alterations in the putamen, which is a key region of the nigrostriatal pathway that has predominantly been implicated in motor function (*Joshua et al., 2009*; *Faure et al., 2005*). On the other hand, tobacco and alcohol use were associated with connectivity alterations in the caudate-NAcc region, which is part of the mesolimbic pathway that has repeatedly been implicated in reward processing and substance use (*Schultz, 2013*; *Wise, 2004*).

Finally, we observed that the change in the second-order mode of connectivity in striatum induced by L-DOPA administration was associated with the change in symptom severity in PD patients. L-DOPA is used as a drug for the treatment of PD, yet not all patients are equally responsive to L-DOPA treatment. When L-DOPA crosses the blood–brain barrier, it is converted into dopamine and is assumed to increase dopaminergic signalling (*Lewitt, 2008*). However, there are differences between PD patients in treatment response, which can be explained by a variety of factors, including differences in the level of systemic L-DOPA uptake from the gut (*Nonnekes et al., 2016*). Our finding thus indicates that the

second-order striatal connectivity mode is differentially sensitive to acute dopaminergic modulation across PD patients and that the amount of this change is associated with the amount of change in symptom severity. This adds to our hypothesis that this mode is associated with dopamine-related functional connectivity and furthermore indicates that studying the dopaminergic system by applying connectopic mapping to resting-state fMRI offers advantages over PET and SPECT scans: PET and SPECT are not only invasive and limited by their low spatial resolution but also depend on indirect measures of dopaminergic signalling such as availability of DaTs and receptors and are therefore not very sensitive to acute, temporal alterations in dopaminergic signalling. In contrast, here we show that connectopic mapping does allow for the investigation of both fine-grained spatial and short-term temporal changes in dopamine-related functional connectivity.

In conclusion, our results provide evidence that the second-order mode of resting-state functional connectivity in striatum is associated with dopaminergic projections and can be developed into a non-invasive biomarker for investigating dopaminergic (dys)function. This may have wide-ranging clinical and scientific applications across disorders associated with dopaminergic functioning. For example, in the diagnostic work-up of movement disorders where DaT SPECT is currently used to distinguish between PD and essential tremor or dystonic tremor, the resting-state fMRI derived second-order connectivity mode might be used instead. The correlation with symptom severity suggests that this mode might also be used as a progression biomarker, for example, to track differences in rate of progression in future intervention studies of new experimental medications aimed at modifying the course of PD. Our results furthermore suggest that this striatal connectivity mode is associated with functions of both the nigrostriatal and mesolimbic pathway, and that it might be possible to differentiate between the two dopaminergic pathways by considering in which striatal subregion that gradient is altered: connectivity alterations seem to occur in putamen for functions associated with the nigrostriatal pathway and in ventral caudate/NAcc for functions associated with the mesolimbic pathway. However, the exact mapping of this striatal connectivity mode on both pathways as well as its relation with the first-order, ventromedial-to-dorsolateral striatal gradient, which we previously linked to goal-directed behaviours, is subject for further investigation.

## Materials and methods

### Resting-state fMRI data of the HCP dataset

For our first analysis, we used resting-state fMRI data from the HCP, an exceptionally high-quality, publicly available neuroimaging dataset (*Van Essen et al., 2013*). HCP participants were scanned on a customized 3 T Siemens Skyra scanner (Siemens AG, Erlanger, Germany) and underwent two sessions of two 14.4 min multiband accelerated (TR = 0.72 s) resting-state fMRI scans with an isotropic spatial resolution of 2 mm. Here, we included participants from the S1200 release who completed at least one resting-state fMRI session (2 × 14.4 min) and for whom data was reconstructed with the r227 reconstruction algorithm. (The reconstruction algorithm was upgraded in late April 2013 from the original 177 ICE version to the 227 upgraded ICE version. As the reconstruction version has been shown to make a notable signature on the data that can make a large difference in fMRI data analysis [for details, see https://wiki.humanconnectome.org/display/PublicData/Ramifications+of+Image+Reconstruction+Version+Differences], we only included participants with r227 reconstructions.) This resulted in the inclusion of 839 participants (aged 22–37 years; 458 females). Resting-state fMRI data were preprocessed according to the HCP minimal processing pipeline (*Glasser et al., 2013*), which included corrections for spatial distortions and head motion, registration to the T1w structural image, resampling to 2 mm MNI152 space, global intensity normalization, and high-pass filtering with a cutoff at 2000s. The data were subsequently denoised using ICA-FIX – an advanced independent component analysis-based artefact removal procedure (*Salimi-Khorshidi et al., 2014*) – and smoothed with a 6 mm kernel.

### Connectopic mapping of the striatum in the HCP dataset

We estimated connection topographies from the HCP resting-state fMRI data using the first session (2 × 14.4 min) for each subject. To this end, we used connectopic mapping (*Haak et al., 2018*), a novel method that enables the dominant modes of functional connectivity change within the striatum to be traced on the basis of the connectivity between each striatal voxel and the

rest of the brain (see *Figure 1*). In previous work, we showed that the dominant mode (zeroth-order mode) of connectivity in the striatum obtained with connectopic mapping represented its anatomical subdivision into putamen, caudate, and NAcc. Since higher-order modes are restricted by lower-order modes, we decided to take the anatomical subdivision in the striatum into account by applying connectopic mapping in the current work to the left and right putamen and caudate-NAcc striatal subregions separately, thereby also increasing regional specificity. When referring to the second-order mode of connectivity in striatum, we thus refer to the combination of the second-order connectivity modes of putamen and caudate-NAcc. We did not apply connectopic mapping to the NAcc and caudate separately as the left NAcc and right NAcc only include 136 voxels and 127 voxels, respectively. We expect that this very small region is too homogenous in terms of connectivity with cortex to estimate reliable overlapping connectivity modes. Masks for the striatal regions were obtained by thresholding the respective regions from the Harvard-Oxford atlas at 25% probability.

In brief, we rearranged the fMRI time-series data from each striatal subregion and all grey-matter voxels outside the striatum into two time-by-voxels matrices. Since the latter is relatively large, we reduced its dimensionality using a lossless singular value decomposition (SVD). We then computed the correlation between the voxel-wise striatal time-series data and the SVD-transformed data from outside the striatum, and subsequently used the $\eta^2$ coefficient to quantify the similarities among the voxel-wise fingerprints (*Haak et al., 2018*). Next, we applied the Laplacian eigenmaps non-linear manifold learning algorithm (*Belkin and Niyogi, 2002*) to the acquired similarity matrix, which resulted in a set of overlapping, but independent, vectors representing the dominant modes of functional connectivity change across striatum (i.e., connection topographies). Note that this can be done at the group level by using the average of the individual similarity matrices or individually for each subject (as used for statistical analysis). For each subject, modes were aligned to the group-level connectivity mode (by inversion if negatively correlated) to enable visual and statistical comparisons across subjects. We selected the second-order striatal connectivity mode (both the group-average and subject-specific modes) for further analyses.

Finally, to enable statistical analysis over these connection topographies, we fitted spatial statistical models to obtain a small number of coefficients summarizing the second-order connectivity mode of each striatal subregion in the X, Y, and Z axes of MNI152 coordinate space. For this, we use 'trend surface modelling' (*Gelfand et al., 2010*), an approach originally developed in the field of geostatistics, but that has wide-ranging applications due to its ability to model the overall distribution of properties throughout space as a simplified surface. Here, we use the TSM approach to predict each individual subject's connection topography by fitting a set of polynomial basis functions defined by the coordinates of each striatal location. We fit these models using Bayesian linear regression (*Bishop, 2006*), where we employed an empirical Bayes approach to set model hyperparameters. Full details are provided elsewhere (*Bishop, 2006*), but this essentially consists of finding the model hyperparameters (controlling the noise- and the data variance) by maximizing the model evidence or marginal likelihood. This was achieved using conjugate gradient optimization. For fixed hyperparameters, the posterior distribution over the trend coefficients can be computed in closed form. This, in turn, enables predictions for unseen data points to be computed. We used the maximum a posteriori estimate of the weight distribution as an indication of the importance of each trend coefficient in further analyses. To select the degree of the interpolating polynomial basis set, we fit these models across polynomials of degree 2–5 and then compared the different model orders using a scree plot analysis (*Cattell, 1966*). This criterion strongly favoured a polynomial of degree 2 for the putamen subregion and a polynomial of degree 4 for the caudate-NAcc subregion. This means that the connectivity mode in putamen was modelled with linear and quadratic functions in the X, Y, and Z directions of MNI152 coordinate space (six TSM coefficients) and the connectivity mode in the caudate-NAcc region with linear, quadratic, cubic, and quartic functions in the X, Y, and Z directions of MNI152 coordinate space (12 TSM coefficients). The TSM coefficients of the fitted polynomial basis functions describe the rate at which the connectivity mode changes along a given spatial dimension and can be used for statistical analysis. The polynomials summarized the connectivity modes well, explaining the following mean ± SD of the variance: left putamen: 90.5% ± 4.16%; right putamen: 90.2% ± 4.64%; left caudate-NAcc: 88.6% ± 2.54%; right caudate-NAcc: 89.4% ± 2.15%.

## DaT SPECT imaging in the PPMI dataset

To determine whether the second-order striatal connectivity mode was associated with dopaminergic projections in striatum, we investigated its spatial correspondence with DaT availability as revealed by DaT SPECT imaging. We selected DaT SPECT scans for all 210 healthy controls (aged 30–84 years; 71 females) included in the PPMI (*Marek et al., 2011*) database (https://www.ppmi-info.org/data). PPMI is a public–private partnership funded by the Michael J. Fox Foundation for Parkinson's Research and funding partners (for up-to-date information, please visit https://www.ppmi-info.org/fundingpartners). Each participating PPMI site obtained ethical approval before study initiation, and written informed consent according to the Declaration of Helsinki was obtained from all participants in the study. PPMI scans were obtained at 24 different sites and acquired with a total of seven different SPECT camera models from different manufacturers. All brain images were registered to MNI152 standard space using a linear affine transformation implemented in FSL FLIRT (*Jenkinson and Smith, 2001*; *Jenkinson et al., 2002*) and a custom DaT SPECT template (http://www.nitrc.org/projects/spmtemplates; *García-Gómez et al., 2013*). Typically, analysis of DaT SPECT images is limited to determining the striatal-binding ratio. This index of DaT availability is calculated by normalizing the average DaT uptake in the striatum (or a striatal subregion) by a reference region of minimal DaT availability (e.g., cerebellum or occipital cortex). However, here we were interested in the detailed spatial profile of DaT availability across the striatum. To obtain this spatial profile, we intensity-normalized all raw DaT SPECT images (*Llera et al., 2019*) so as to optimize contrast in the DaT SPECT image and take into account variability in the DaT SPECT scans across the PPMI dataset as a result of different cameras and different scan sites. Finally, we averaged across all subjects and masked the striatum to obtain the average DaT SPECT image of the striatum.

## Mapping the second-order striatal connectivity mode onto DaT availability

Next, we quantified the similarity between the second-order mode of connectivity in striatum and the DaT SPECT image. To this end, we combined the average (i.e., group level) second-order connectivity modes of putamen and caudate-NAcc obtained in the high-resolution HCP dataset and computed the voxel-wise spatial correlation of this mode with the average (i.e., group level) DaT SPECT image of striatum obtained in the PPMI dataset. Given that the voxel-wise spatial correlation between both images might be inflated due to potential spatial autocorrelation effects (i.e., the images represent smooth spatial functions), we additionally computed the correlation between the TSM coefficients modelling the group-average connectivity mode and the group-average DaT SPECT scan since TSM coefficients are orthogonal. More specifically, the TSM coefficients modelling the left and right putamen ($2 \times 6$) and caudate-NAcc regions ($2 \times 12$) were combined and the correlation was computed across all these 36 TSM coefficients. In addition, to show that our results do not heavily depend on the chosen model order, we repeated this analysis using model order 3 (i.e., a cubic model with nine TSM coefficients) for both the putamen and caudate-NAcc regions (i.e., $4 \times 9 = 36$ TSM coefficients).

For a subsample of PPMI participants with a DaT SPECT scan, there was also a low-resolution resting-state fMRI scan available (130 datasets from PD patients and 14 from controls). We therefore also investigated the *within-subject* spatial correspondence between the DaT SPECT scan and the second-order connectivity mode for these subjects in the PPMI dataset. This procedure is detailed in Appendix 1—Supplementary analyses.

## Resting-state fMRI data of the PD dataset

Given that PD is characterized by a loss of dopaminergic neurons (*Fearnley and Lees, 1991*), we investigated whether the second-order striatal connectivity mode was altered in PD. For this analysis, we used high-resolution resting-state fMRI data from a cohort consisting of 39 patients with PD (aged 38–81, 23 females) and 20 controls (aged 42–80, 9 females), recruited at the Centre of Expertise for Parkinson & Movement Disorders at the Radboud University Medical Center (Radboudumc) in Nijmegen and scanned at the Donders Institute in Nijmegen, the Netherlands (*Dirkx et al., 2019*). All patients were diagnosed with idiopathic PD (according to the UK Brain Bank criteria), and all patients had a mild to severe resting tremor besides bradykinesia. In 20 patients, the motor symptoms were right-dominant, in 19 patients left-dominant (dominance here refers to the side of the body displaying the most prominent motor symptoms [including tremor]), which is believed to correspond with a

dopamine depletion dominant to contralateral hemisphere in the brain. The study was approved by the local ethics committee, and written informed consent according to the Declaration of Helsinki was obtained from all participants. Detailed sample characteristics can be found in *Table 2*.

Patients with PD underwent two 10 min resting-state fMRI sessions, that is, a placebo session and L-DOPA session, separated by at least a day on a 3 T Siemens Magnetom Prisma^fit scanner. Resting-state fMRI scans were obtained with an interleaved high-resolution multiband sequence (TR = 0.860 s, voxel size = 2.2 mm isotropic, TE = 34 ms, flip angle = 20°, 44 axial slices, multiband acceleration factor = 4, volumes = 700). Under both conditions, patients were scanned after over-night fasting in a practically defined off state, that is, more than 12 hr after intake of their last dose of dopaminergic medication. During one session, patients were scanned after administration of L-DOPA, that is, they received a standardized dose of 200/50 mg dispersible levodopa/benserazide. During the other session, patients received placebo (cellulose powder). The cellulose powder and L-DOPA/benserazide were dissolved in water and therefore undistinguishable (visually and olfactory) for the participants as confirmed by a pilot study. Patients also received 10 mg domperidone to improve gastrointestinal absorption of levodopa and reduce side effects. The order of sessions was counterbalanced and the resting-state fMRI scan started on average 48 min (range: 25–70 min) after taking L-DOPA or placebo. Symptom severity was assessed during both sessions with part III (assessment of motor function by a clinician) of the Movement Disorder Society UPDRS (*Goetz et al., 2008*), and an electromyogram (EMG) of the hand was recorded to monitor tremor-related activity. In light of ethical considerations, control participants did not receive L-DOPA and placebo, they just underwent two typical resting-state fMRI sessions during which the UPDRS was not administered.

Preprocessing of the resting-state fMRI data included removal of the first five volumes to allow for signal equilibration, primary head motion correction via realignment to the middle volume MCFLIRT (*Jenkinson et al., 2002*), grand mean scaling, and spatial smoothing with a 6 mm FWHM Gaussian kernel. The preprocessing pipeline was furthermore designed to rigorously correct for potential tremor-induced head motion-related artefacts. To this end, we used ICA-AROMA (*Pruim et al., 2015*), an advanced ICA-based motion correction procedure to identify and remove secondary head motion-related artefacts with high accuracy while preserving signal of interest (*Pruim et al., 2015*; *Parkes et al., 2018*). Next, any remaining motion artefacts were removed from the data by regressing out the EMG parameters in addition to the white matter and CSF signal (*Helmich et al., 2012*). Finally, the data were temporally filtered with a high-pass filter of 0.01 Hz before being resampled to 2 mm MNI152 space.

## Investigating the second-order striatal mode in PD

We applied connectopic mapping to the preprocessed resting-state fMRI data of each session from every participant and selected the second-order connectivity mode for further analyses using the same procedure as in the HCP dataset. The subject-specific second-order striatal connectivity modes for control participants were again consistent across the two fMRI sessions mean ± SD $\rho$ = 0.85 ± 0.11 (individual subregions: left putamen: $\rho$ = 0.78 ± 0.10; right putamen: $\rho$ = 0.82 ± 0.12; left caudate-NAcc: $\rho$ = 0.87 ± 0.14; and right caudate-NAcc: $\rho$ = 0.92 ± 0.08). The polynomials also summarized the connectivity modes well, explaining mean ± SD 78.6% ± 11.8% of the variance across the striatum in controls (individual subregions: left putamen: 67.9% ± 16.2%; right putamen: 65.3% ± 21.2%; left caudate-NAcc: 90.5% ± 4.28%; right caudate-NAcc: 90.8% ± 5.63%), and explaining mean ± SD 78.0% ± 10.5% of the variance across striatum in PD patients under placebo (individual subregions: left putamen: 63.6% ± 19.6%; right putamen: 69.5% ± 13.0%; left caudate-NAcc: 88.5% ± 4.69%; right caudate-NAcc: 90.4% ± 4.58%). While these numbers are lower than observed for the connectivity modes obtained from the HCP dataset – which is not surprising given the exceptionally high quality of the HCP dataset – the reproducibility and explained variance of the TSM coefficients are still substantial.

We conducted four different analyses. All these analyses were conducted separately for the left- and right-dominant PD groups – given that the dopamine depletion is dominant to different hemispheres in these two groups – and separately for the putamen and caudate-NAcc subregions (left + right hemisphere combined) to increase regional specificity as PD is known to affect the putamen region of the striatum before the caudate-NAcc region. In all these analyses, we therefore corrected

for multiple comparisons using a Bonferroni-corrected $\alpha$-level of 0.0125 (0.05/4 (2 patient groups * 2 subregions)).

In our first analysis, we compared the second-order striatal connectivity mode between the control and PD groups. To this end, we conducted omnibus tests comparing the TSM coefficients modelling the second-order striatal gradient of the placebo session in the PD group with the TSM coefficients modelling the striatal gradient of the first session of the control group. More specifically, group differences in the TSM coefficients were assessed by using a likelihood ratio test in the context of a logistic regression. We report the $X^2$ (likelihood test) and corresponding p-value of tests that revealed significant group differences. Second, since PD is a heterogeneous disease, we also conducted an analysis taking this variability into account by investigating associations between the TSM coefficients modelling the second-order striatal gradient and symptom severity in PD patients. To this end, we conducted GLMs that included the TSM coefficients modelling the gradient during the placebo session to predict UPDRS symptom severity scores. For all identified associations with UPDRS symptom severity, we report the $X^2$ (likelihood test) and the corresponding p-values, and post-hoc compute Pearson correlations between UPDRS symptom severity and the individual TSM coefficients to determine which coefficients most strongly contributed to the effect.

Third, we assessed differences in the second-order striatal connectivity mode between the placebo and L-DOPA session in both PD groups. More specifically, session differences in the TSM coefficients were assessed by using a likelihood ratio test in the context of a logistic regression. We report the $X^2$ (likelihood test) and corresponding p-value of tests that revealed significant differences between the placebo and L-DOPA session. Finally, treatment response to L-DOPA is known to differ among patients with PD. To take this variability across patients into account, we also investigated whether the L-DOPA-induced change in the second-order striatal connectivity mode was associated with L-DOPA-induced changes in UPDRS symptom severity. More specifically, we calculated the difference in UPDRS symptom scores and TSM coefficients between the placebo and L-DOPA session and investigated associations within the GLM framework. For all identified associations, we post-hoc computed Pearson correlations between the change in UPDRS symptom severity and the change in individual TSM coefficients to determine which coefficients most strongly contributed to the effect.

## Investigating the second-order striatal mode in relation to tobacco and alcohol use

Given that alterations in dopaminergic functioning have also been implicated in substance use, we also investigated the association between the second-order striatal connectivity mode and tobacco use as well as alcohol use across high users within the HCP dataset. To this end, we selected HCP participants testing negative for acute drug and alcohol use but who reported to have consumed ≥3 light and/or ≥1 heavy alcoholic drinks per day over the past week (N = 30), and participants reporting to have smoked ≥5 cigarettes every day over the past week (N = 38). Effects of smoking and drinking were analysed separately, and we again modelled the second-order striatal connectivity mode separately for left and right putamen and left and right caudate-NAcc. We conducted GLMs that included next to the amount of use over the past week (i.e., the total number of alcohol drinks or the total number of times tobacco was smoked), the TSM coefficients modelling the second-order striatal connectivity mode (i.e., the TSM coefficients modelling the putamen or caudate-NAcc). Multiple comparison correction was applied using a Bonferroni-corrected $\alpha$-level of 0.0125 (2 substances * 2 striatal subregions). For all identified associations with the amount of alcohol use or tobacco use, we report the $X^2$ (likelihood test) and the corresponding p-values and post-hoc compute Pearson correlations between the amount of use and the individual TSM coefficients to determine which coefficients most strongly contributed to the association.

## Supplementary analyses

To further demonstrate the high specificity of the second-order connectivity mode to the DaT SPECT scan over and above other PET scans (i.e., other neurotransmitter systems), we computed correlations with the TSM coefficients of all PET scans, tapping into various neurotransmitter systems, included in the publicly available JuSpace toolbox (*Dukart et al., 2021*). This analysis is described in Appendix 1—Supplementary analyses. We also investigated whether the mapping of the second-order connectivity mode onto DaT SPECT scan was influenced by residual head motion. Furthermore, for the other

analyses described above (effects of diagnosis and L-DOPA in the PD dataset and associations with tobacco and alcohol use in the HCP dataset), we conducted post-hoc sensitivity analyses to rule out that the group differences and associations revealed by our analyses were dependent on age and sex. In addition, we investigated whether the associations with the amount of tobacco and alcohol use persisted under different usage thresholds. All these analyses are also described in Appendix 1—Supplementary analyses.

## Acknowledgements

We made use of HCP data that were provided by the Human Connectome Project, WU-Minn Consortium (principal investigators: David Van Essen and Kamil Ugurbil; 1U54MH091657) funded by the 16 NIH Institutes and Centres that support the NIH Blueprint for Neuroscience Research; and by the McDonnell Centre for Systems Neuroscience at Washington University. We also used data from the Parkinson's Progression Markers Initiative (PPMI) database (https://www.ppmi-info.org/data). PPMI – a public–private partnership – is funded by the Michael J Fox Foundation for Parkinson's Research funding partners Abbvie, Avid Radiopharmaceuticals, Biogen Idec, BioLegend, Bristol- Myers Squibb, Eli Lilly & Co., F Hoffman-La Roche, Ltd., GE Healthcare, Genentech, GlaxoSmithKline, Lundbeck, Merck, MesoScale Discovery, Piramal, Pfizer, Sanofi Genzyme, Servier, Takeda, Teva, and UCB. We further used PET scans available in the JuSpace toolbox: https://github.com/juryxy/JuSpace (Dukart et al., 2021).

## Additional information

### Competing interests

Christian F Beckmann: C.F.B. is director and shareholder in SBGneuro Ltd. The other authors declare that no competing interests exist.

### Funding

| Funder | Grant reference number | Author |
| --- | --- | --- |
| Nederlandse Organisatie voor Wetenschappelijk Onderzoek | Vidi Grant No. 864-12-004 | Christian F Beckmann |
| Nederlandse Organisatie voor Wetenschappelijk Onderzoek | Vici Grant No. 17854 | Christian F Beckmann |
| Nederlandse Organisatie voor Wetenschappelijk Onderzoek | Vidi Grant No. 016.156.415 | Andre F Marquand |
| Nederlandse Organisatie voor Wetenschappelijk Onderzoek | Veni Grant No. 016.171.068 | Koen V Haak |
| Nederlandse Organisatie voor Wetenschappelijk Onderzoek | Veni Grant No. 91617077 | Rick Helmich |
| Nederlandse Organisatie voor Wetenschappelijk Onderzoek | NWO-CAS Grant No. 012-200-013 | Christian F Beckmann |
| ZonMw | Rubicon Grant No. 452172019 | Marianne Oldehinkel |
| Dutch Brain Foundation | Grant F2013(10–15) | Rick C Helmich |

The funders had no role in study design, data collection and interpretation, or the decision to submit the work for publication.

## Author contributions

Marianne Oldehinkel, Conceptualization, Data curation, Formal analysis, Investigation, Methodology, Validation, Visualization, Writing - original draft, Writing – review and editing; Alberto Llera, Conceptualization, Formal analysis, Resources, Writing – review and editing; Myrthe Faber, Methodology, Writing – review and editing; Ismael Huertas, Formal analysis, Methodology, Resources, Writing – review and editing; Jan K Buitelaar, Bastiaan R Bloem, Writing – review and editing; Andre F Marquand, Methodology, Software, Writing – review and editing; Rick C Helmich, Funding acquisition, Resources, Writing – review and editing; Koen V Haak, Formal analysis, Investigation, Methodology, Software, Supervision, Writing – review and editing; Christian F Beckmann, Funding acquisition, Investigation, Methodology, Resources, Supervision, Writing – review and editing

## Author ORCIDs

Marianne Oldehinkel (D) http://orcid.org/0000-0002-7573-0548
Alberto Llera (D) http://orcid.org/0000-0002-8358-8625
Andre F Marquand (D) http://orcid.org/0000-0001-5903-203X

## Ethics

Clinical trial registration https://www.trialregister.nl/trial/4940.
All participants from whom data was used in this manuscript, provided written informed consent (and consent to publish) according to the declaration of Helsinki. For the HCP dataset ethical approval was given by the Washington University Institutional Review Board (IRB), for the PPMI dataset ethical approval was obtained locally at each of the participating sites, and for our local PD dataset ethical approval was obtained from the local ethical committee (Commissie Mensgebonden Onderzoek MO Arnhem Nijmegen, CMO 2014/014).

## Decision letter and Author response

Decision letter https://doi.org/10.7554/eLife.71846.sa1
Author response https://doi.org/10.7554/eLife.71846.sa2

# Additional files

## Supplementary files

• Transparent reporting form

## Data availability

We made use of publicly available data from the Human Connectome Project (HCP) dataset, from publicly available data from the Parkinson Progression Marker Initiative (PPMI) dataset, and from a local PD dataset that was part of a clinical trial. See https://www.humanconnectome.org/study/hcp-young-adult/document/1200-subjects-data-release for access to the HCP data. The subject identifiers from the HCP dataset used in our first analysis can be found in Appendix 2 - Table 4. Please note that the subject identifiers from the subset of HCP subjects included in the nicotine-use and alcohol-use analyses cannot be provided, since the access to information about substance use is restricted. For more information about applying to get access to the HCP restricted data and for the HCP restricted data use terms see: https://www.humanconnectome.org/study/hcp-young-adult/document/wu-minn-hcp-consortium-restricted-data-use-terms. For access to the PPMI dataset, see https://www.ppmi-info.org/access-data-specimens/download-data. The subject identifiers from the PPMI dataset used in our analyses can be found in Appendix 2 - Tables 5 and 6. All derived and anonymized individual data from our local PD dataset are available at the Donders Repository: https://data.donders.ru.nl/. The code used for the connectopic mapping procedure in all three datasets is available at the following Github repository: https://github.com/koenhaak/congrads. In addition, for supplementary analyses, we further used PET scans available in the JuSpace Toolbox: https://github.com/juryxy/JuSpace.

The following previously published datasets were used:

| Author(s) | Year | Dataset title | Dataset URL | Database and Identifier |
|-----------|------|---------------|-------------|-------------------------|
| Van Essen DC, Smith SM, Barch DM, Behrens TE, Yacoub E, Ugurbil K | 2013 | The WU-Minn human connectome project: an overview | https://www.humanconnectome.org/study/hcp-young-adult/document/1200-subjects-data-release | Human Connectome Project, Appendix2-Table4 |
| Marek K, Jennings D, Lasch S, Siderowf A, Tanner C, Simuni T | 2011 | The Parkinson Progression Marker Initiative (PPMI) | https://www.ppmi-info.org/access-data-specimens/download-data | Parkinson Progression Marker Initiative, Appendix2-Tables-5-6 |
| Dukart J, Holiga S, Rullmann M, Lanzenberger R, Hawkins PC, Mehta MA | 2021 | JuSpace toolbox | https://github.com/juryxy/JuSpace | Github, JuSpace |

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

## Appendix 1

### Supplementary analyses

Post-hoc analyses comparing the second-order striatal connectivity mode with PET markers of other neurotransmitter systems

We investigated the spatial correspondence of the second-order connectivity mode to multiple PET markers indexing various neuromodulatory systems. To this end, we made use of various PET scans tapping into different neurotransmitter systems (group-averages of 11–36 controls) implemented in the publicly available JuSpace toolbox (https://github.com/juryxy/JuSpace). The included PET scans for the serotonin system are 5HT1a receptor (5HT1a_WAY), 5HT1b receptor (5HT1b_P943), 5HT2a receptor (5HT2a_ALT), and the serotonin transporter (SERT_DASB and SERT_MADAM). The included PET scans for the dopamine system (other than DaT SPECT) are dopamine type 1 receptor (D1_SCH23390), dopamine type 2 receptor (D2_RACLOPRIDE), and FDOPA (FDOPA_f18). In addition, the following neurotransmitter systems were also included: for GABA the GABAa receptor (GABAa_FLUMAZENIL), for noradrenalin the noradrenalin receptor (NAT_MRB), and for the opiod system the mu opiod receptor (MU_CARFENTANIL).

We applied TSM to each of these PET scans in the striatum (as in the main analysis, the rest of the brain was masked and not included) and computed correlations between TSM coefficients modelling the second-order connectivity mode and the TSM coefficients modelling these PET-derived markers. For each PET scan, the correlations with the TSM coefficients were normalized using the Fisher r-to-z transformation and the absolute correlation was taken. These normalized correlations are visualized in the top panel of *Figure 2—figure supplement 1*. As can be observed, the correlation of the second-order connectivity mode with the DaT SPECT scan is substantially higher than that of any other PET marker. Though it is noticeable that some of the correlations with the PET markers for the serotonin system (SERT_DASB transporter and 5HT1b receptor) that are also known to have a high density in the striatum are also relatively high. Nevertheless, these markers only reach about half of the correlation value of DaT SPECT. To support the robustness and significance of the correlation of our second-order connectivity mode with the DaT SPECT scan over and above the correlation with the markers of the serotonergic system, we tested the correlation between the TSM coefficients obtained for the second-order connectivity mode and the DaT SPECT scan in striatum for significance using permutation testing (N = 10,000). More specifically, we permuted corresponding TSM coefficients obtained for each of the PET markers and thereby generated a null distribution by computing the absolute (Fisher r-to-z normalized) correlations between the connectivity mode TSM coefficients and the permuted TSM coefficients of the other PET markers. Permutations were conducted separately for each coefficient, not permuting across coefficients to ensure interchangeability under the null assumption of no differentiation across different PET markers.

As can be observed in the bottom panel of *Figure 2—figure supplement 1*, all permuted correlations are lower than the correlation observed between the DaT SPECT scan and the connectivity mode, indicating that the observed correlation between the connectivity mode and DaT SPECT scan is highly significant and unlikely to be obtained by chance. Furthermore, using this null distribution, we defined the Bonferroni-corrected threshold for significance corresponding to p=0.0008 (i.e., p=0.01/12 PET and SPECT scans), which we added to the top figure displaying the correlations with the other PET tracers. This not only confirms that our results are highly significant, but also that the correlations obtained for the other PET markers, including those of the serotonin system, are not only substantially lower, but also do not pass the threshold for significance based on the null distribution. Of note, the correlation with other markers of the dopamine system, such as the D1 and D2 receptor, as opposed to DaT SPECT is not particularly high. However, this is not surprising since these receptors are present on postsynaptic neurons and are likely representative of postsynaptic dopaminergic projections from striatum to cortex, rather than the presynaptic dopaminergic projections from the midbrain to the striatum reflected by the DaT SPECT scan.

### Within-subject correspondence between the second-order striatal connectivity mode and the DaT SPECT scan

In the article, we demonstrated that the second-order striatal connectivity mode at the group level (obtained by averaging this mode across all 839 HCP subjects) showed a very high spatial

correlation (*r* = 0.884) with the group-level DaT SPECT image of striatum (obtained by averaging the DaT SPECT images across all 209 PPMI controls). We also aimed to demonstrate that this mapping can be replicated at the within-subject level by investigating the within-subject spatial correspondence between this connectivity mode and the DaT SPECT scan acquired in the PPMI dataset. However, while the PPMI dataset has resting-state fMRI data available for a small subsample of its participants (14 controls with one resting-state fMRI dataset each and 82 PD patients with 130 resting-state fMRI datasets combined; in case of multiple assessments per subject they were separated by at least 1 year), it is of a relatively low temporal and spatial resolution (TR = 2400 ms, 210 time points, 3.3 mm isotropic resolution) compared to the HCP data (TR = 720 ms, 2,400 time points, 2.0 mm isotropic resolution). While this resolution is sufficient for typical resting-state fMRI analyses, the precise delineation of the very fine-grained and overlapping connectivity modes using connectopic mapping calls for high-resolution data. The single-subject connectivity modes in the PPMI dataset (as opposed to the HCP single-subject modes and group-level modes) might therefore not be of sufficient quality and reliable for every subject. To address this issue, we first computed the spatial correlation of each subject's individual connectivity mode with that of the group-average HCP connectivity mode as well as with the DaT SPECT scan of each subject (see *Figure 3—figure supplement 1*). In this analysis, the second-order striatal connectivity mode was modelled separately (and correlations were calculated separately) for the left and right putamen and caudate-NAcc subregions. This revealed highly significant positive correlations (0.68 > *r* < 0.91, all p<4.0e21) across both controls and patients, suggesting that if the connectivity mode of a subject resembles the HCP group-average connectivity mode – assumed to be an index of good quality – a high spatial similarity can be observed between the connectivity mode and the DaT SPECT scan of that subject. Next, we selected those subjects with good-quality connectivity modes as determined by a spatial correlation of *r* > 0.5 with the group-average connectivity mode in the HCP dataset. Within this sample of 73–86 datasets from PD patients and 6–8 datasets from controls (dependent on the striatal subregion), we not only replicated the spatial correspondence between the connectivity mode and DaT SPECT scan at the group level (patients: *r* = 0.714; control group: *r* = 0.721) but also observed significant *within-subject* spatial correlations (0.44 > *r* < 0.63; mean = 0.58, 95% CI = [0.56,0.60]) between the connectivity mode and DaT SPECT scan (see *Figure 3*). While we were able to replicate the spatial correlation between the second connectivity mode and the DaT SPECT scan at the *within-subject* level, this correlation (*r* = 0.58) is not as high as the spatial correlations observed in the group level (i.e., *r* = 0.721 and *r* = 0.714 for PPMI controls and PD patients, respectively, and *r* = 0.884 between the DaT SPECT scan in PPMI controls and the connectivity mode in HCP participants). This is, however, not surprising given the relatively low temporal and spatial resolution of the resting-state fMRI scan of the PPMI dataset. However, to our knowledge, there is currently no dataset available that includes both a high-resolution resting-state fMRI scan and a DaT SPECT scan from the same participants.

## Post-hoc analyses of head motion

To demonstrate that the mapping of the group-average second-order connectivity mode onto the group-average DaT SPECT scan (*r* = 0.884) was not influenced by residual head motion, we generated connectivity modes for the 10% highest movers (N = 84, meanFD range: 0.0376–0.0538) and 10% lowest movers (N = 84, meanFD range: 0.1354–0.3155) of the HCP dataset and computed the spatial correlation to the group-average DaT SPECT scan (N = 209, no head motion metrics were available, so we used the entire sample). This analysis revealed very similar connectivity modes (see *Figure 3—figure supplement 2*) and a very similar spatial correlation for the low FD group (*r* = 0.883) and the high FD group (*r* = 0.886), indicating that this mapping was not induced by residual head motion.

Next, we aimed to demonstrate that the mapping between the connectivity mode and the DaT SPECT scan at the within-subject level was also not influenced by residual head motion. To this end, we divided the subsample of the PPMI dataset where both a resting-state fMRI scan and DaT SPECT are available (used for mapping the second-order connectivity mode onto the DaT SPECT at the within-subject level) in half based on the meanFD (*meanFD cutoff = 0.126*) and computed the within-subject correlation between the DaT SPECT and connectivity mode for the 'low motion' and 'high motion' halves of the sample separately. This analysis revealed that the within-subject spatial correlation between the DaT SPECT and second-order connectivity mode was virtually identical between the low and high meanFD samples (see *Figure 3—figure supplement 2*). Although it does appear that for the caudate-NAcc region the correlations between the connectivity mode and the DaT SPECT scan are slightly lower for the high motion half than for the

low motion half. This indicates that the high spatial correlation between the connectivity mode with the DaT SPECT scan is not artificially induced by head motion, but that residual head motion instead weakens this correlation. All these additional analyses thus suggest that the proposed biomarker is not picking up on residual motion.

## Post-hoc analyses of age and sex

For all the analyses described in the article (effects of diagnosis and L-DOPA in the PD dataset and associations with smoking and drinking in the HCP dataset), we conducted post-hoc sensitivity analyses to rule out that the group differences and behavioural associations revealed by our analyses were dependent on age and sex. To this end, we conducted two types of analyses. First, we repeated our main analyses by including covariates for age and sex in our statistical models in addition to the TSM coefficients to verify that effects remained (close to) significant when including these demographic variables. Next, we only included age and sex in our statistical models (without the TSM coefficients) to verify that effects were not explained by age and/or sex only. The outcomes of these analyses ($X^2$ and p-value) are listed in *Appendix 2—table 1* and demonstrate that none of the significant effects observed in our main analyses were dependent on age or sex. However, adding age and sex (age in particular) did increase the significance of findings substantially for the analyses investigating the L-DOPA-induced changes. This might be explained by the fact that patients who are older often have more severe PD and do not benefit as much anymore from L-DOPA treatment.

## Post-hoc analyses using different usage thresholds for tobacco and alcohol use

We also investigated whether the associations of the second-order mode of connectivity in striatum with the amount of tobacco use and alcohol use persisted under different usage thresholds. For both tobacco and alcohol use, we chose a daily usage threshold lower (≥2× tobacco/≥1× alcoholic drink) and a daily usage threshold higher (≥8× tobacco/≥3× alcoholic drink) than the one used in the main analysis (≥5× tobacco/≥3× light alcoholic and/or ≥1× hard liquor drinks a day). Please note that the aim of these analyses is not necessarily to show that effects remain significant as under different usage thresholds the sample size and statistical power will change, but rather that the explained variance remains high. Nevertheless, apart from the low-usage threshold for alcohol use, all effects also remained significant, as can be observed in *Appendix 2—table 2* and *Appendix 2—table 3*, indicating that the associations with tobacco and alcohol use were not specific to the chosen usage threshold. However, a pattern that is visible is that associations become stronger when only including the highest users in this population-based sample in the analysis.

# Appendix 2

## Supplementary tables

**Appendix 2—table 1.** Post-hoc analyses of age and sex.

| | Original analysis | | Original analysis + age and sex | | Age and sex only | |
|---|---|---|---|---|---|---|
| | $X^2$ | p-Value | $X^2$ | p-Value | $X^2$ | p-Value |
| *Putamen* | | | | | | |
| **Patients vs. controls** Right tremor-dominant Parkinson's disease | 27.17 | 0.007 | 27.21 | 0.018 | 0.48 | 0.786 |
| **UPDRS symptom severity** Right tremor-dominant Parkinson's disease | 22.28 | 0.035 | 23.46 | 0.053 | 2.38 | 0.305 |
| **L-DOPA-placebo difference** Left tremor-dominant Parkinson's disease | 34.07 | 0.001 | 46.14 | <0.001 | 2.42 | 0.299 |
| **L-DOPA-placebo difference** Right tremor-dominant Parkinson's disease | 25.48 | 0.012 | 37.53 | 0.001 | 7.18 | 0.028 |
| *Caudate-NAcc* | | | | | | |
| **Tobacco use** HCP dataset | 49.55 | 0.002 | 53.56 | 0.001 | 1.04 | 0.594 |
| **Alcohol use** HCP dataset | 64.45 | <0.001 | 174.87 | <0.001 | 9.26 | 0.010 |

UPDRS = Unified Parkinson's Disease Rating Scale; L-DOPA = levodopa-benserazide; HCP = Human Connectome Project; NAcc = nucleus accumbens.

**Appendix 2—table 2.** Post-hoc analyses using different thresholds for tobacco use.

| | Original analysis: ≥5× tobacco use a day N = 38 | | ≥2× tobacco use a day N = 62 | | ≥8× tobacco use a day N = 30 | |
|---|---|---|---|---|---|---|
| | $X^2$ | p-Value | $X^2$ | p-Value | $X^2$ | p-Value |
| **Tobacco use** HCP dataset *caudate-NAcc* | 49.55 | 0.002 | 37.96 | 0.035 | 70.54 | <0.001 |

HCP = Human Connectome Project; NAcc = nucleus accumbens.

**Appendix 2—table 3.** Post-hoc analyses using different thresholds for alcohol use.

| | Original analysis: ≥3× light alcoholic and/or ≥1× hard liquor drinks a day N = 30 | | ≥1× alcoholic drinks a day (light and/or hard liquor) N = 103 | | ≥3× alcoholic drinks a day (light and/or hard liquor) * N = 26 | |
|---|---|---|---|---|---|---|
| | $X^2$ | p-Value | $X^2$ | p-Value | $X^2$ | p-Value |
| **Alcohol use** HCP dataset *caudate-NAcc* | 64.45 | <0.001 | 29.94 | 0.187 | 196.57 | <0.001 |

HCP = Human Connectome Project; NAcc = nucleus accumbens.

**Appendix 2—table 4.** Subject IDs from the 839 HCP subjects used in the connectopic mapping analysis.

| | | | | | | | |
|---|---|---|---|---|---|---|---|
| 100206 | 129129 | 155635 | 181636 | 212823 | 385450 | 580044 | 784565 |
| 100610 | 129331 | 155938 | 182032 | 213017 | 386250 | 580347 | 788674 |
| 101006 | 129533 | 156031 | 182436 | 213421 | 387959 | 580650 | 789373 |

*Appendix 2—table 4 Continued on next page*

*Appendix 2—table 4 Continued*

| | | | | | | | |
|---|---|---|---|---|---|---|---|
| 101107 | 129634 | 156435 | 183034 | 213522 | 389357 | 580751 | 792766 |
| 101309 | 129937 | 156536 | 183337 | 214524 | 391748 | 581450 | 792867 |
| 101410 | 130114 | 157437 | 183741 | 214625 | 392447 | 583858 | 793465 |
| 101915 | 130316 | 157942 | 185341 | 214726 | 392750 | 585256 | 800941 |
| 102008 | 130417 | 158136 | 185442 | 217126 | 393247 | 587664 | 802844 |
| 102109 | 130619 | 158338 | 185846 | 219231 | 393550 | 588565 | 803240 |
| 102311 | 130720 | 158843 | 185947 | 220721 | 394956 | 589567 | 804646 |
| 102513 | 130821 | 159138 | 186040 | 221218 | 395251 | 590047 | 809252 |
| 102614 | 131217 | 159340 | 186141 | 223929 | 395756 | 592455 | 810439 |
| 102715 | 131419 | 159441 | 186545 | 227432 | 395958 | 594156 | 810843 |
| 103010 | 131722 | 159744 | 186848 | 227533 | 397154 | 597869 | 812746 |
| 103111 | 131823 | 159845 | 187143 | 228434 | 397861 | 599065 | 814548 |
| 103212 | 132017 | 159946 | 187345 | 231928 | 406432 | 599469 | 814649 |
| 104012 | 132118 | 160729 | 187547 | 233326 | 406836 | 599671 | 815247 |
| 104416 | 133019 | 160830 | 187850 | 236130 | 412528 | 601127 | 816653 |
| 104820 | 134021 | 160931 | 188145 | 237334 | 413934 | 604537 | 818455 |
| 105014 | 134223 | 161630 | 188347 | 238033 | 419239 | 609143 | 818859 |
| 105620 | 134425 | 161832 | 188448 | 239136 | 421226 | 611938 | 820745 |
| 105923 | 134627 | 162026 | 188549 | 248339 | 422632 | 613235 | 822244 |
| 106016 | 134829 | 162228 | 188751 | 250932 | 424939 | 613538 | 825048 |
| 106521 | 135124 | 162733 | 189349 | 255740 | 432332 | 615441 | 825553 |
| 106824 | 135225 | 162935 | 189450 | 256540 | 436239 | 615744 | 825654 |
| 107018 | 135528 | 163129 | 191033 | 257542 | 436845 | 616645 | 826454 |
| 107220 | 135629 | 163331 | 191235 | 257845 | 441939 | 617748 | 827052 |
| 107321 | 135730 | 163836 | 191336 | 257946 | 445543 | 618952 | 828862 |
| 107422 | 136126 | 164030 | 191841 | 263436 | 449753 | 620434 | 832651 |
| 107725 | 136227 | 164131 | 191942 | 268749 | 453441 | 622236 | 833148 |
| 108020 | 136631 | 164636 | 192035 | 268850 | 453542 | 623137 | 833249 |
| 108121 | 136732 | 164939 | 192136 | 270332 | 454140 | 623844 | 835657 |
| 108222 | 137027 | 165032 | 192237 | 274542 | 456346 | 626648 | 837560 |
| 108323 | 137229 | 165234 | 192641 | 275645 | 459453 | 627852 | 837964 |
| 108525 | 137431 | 165436 | 192843 | 280739 | 461743 | 633847 | 841349 |
| 108828 | 137532 | 165638 | 193441 | 280941 | 463040 | 634748 | 843151 |
| 109123 | 137633 | 165941 | 193845 | 281135 | 467351 | 635245 | 844961 |
| 109325 | 137936 | 166438 | 194443 | 283543 | 468050 | 644044 | 845458 |
| 109830 | 138130 | 166640 | 194645 | 285345 | 473952 | 645450 | 849264 |
| 110007 | 138332 | 167036 | 194746 | 285446 | 475855 | 647858 | 849971 |
| 111211 | 138837 | 167238 | 194847 | 286347 | 479762 | 654350 | 852455 |
| 111413 | 139233 | 167440 | 195041 | 286650 | 480141 | 654552 | 856463 |

*Appendix 2—table 4 Continued*

| | | | | | | | |
|---|---|---|---|---|---|---|---|
| 112112 | 139435 | 168240 | 195445 | 287248 | 481042 | 656253 | 856968 |
| 112314 | 139839 | 168341 | 195950 | 289555 | 481951 | 657659 | 867468 |
| 112516 | 140117 | 168745 | 196346 | 290136 | 486759 | 660951 | 869472 |
| 112920 | 140319 | 168947 | 196851 | 295146 | 492754 | 662551 | 870861 |
| 113316 | 140824 | 169040 | 196952 | 297655 | 495255 | 663755 | 871762 |
| 113922 | 140925 | 169444 | 197348 | 298455 | 497865 | 664757 | 872562 |
| 114116 | 141119 | 169545 | 197651 | 299154 | 500222 | 667056 | 873968 |
| 114217 | 141422 | 169747 | 198047 | 299760 | 506234 | 668361 | 877269 |
| 114318 | 141826 | 169949 | 198249 | 300618 | 510225 | 671855 | 878776 |
| 114419 | 142424 | 170631 | 198350 | 300719 | 510326 | 673455 | 878877 |
| 114621 | 143224 | 170934 | 198653 | 303119 | 512835 | 675661 | 880157 |
| 114823 | 143426 | 171128 | 198855 | 303624 | 513130 | 677766 | 882161 |
| 115017 | 144125 | 171330 | 199352 | 304727 | 513736 | 679568 | 884064 |
| 115219 | 144731 | 171431 | 199453 | 305830 | 516742 | 679770 | 886674 |
| 115724 | 144832 | 171532 | 200008 | 308129 | 517239 | 680250 | 888678 |
| 115825 | 144933 | 171633 | 200109 | 308331 | 518746 | 680452 | 891667 |
| 116221 | 145127 | 171734 | 200311 | 309636 | 519647 | 683256 | 894067 |
| 116423 | 145531 | 172029 | 200513 | 310621 | 519950 | 686969 | 894774 |
| 116524 | 145632 | 172130 | 200917 | 311320 | 520228 | 687163 | 898176 |
| 116726 | 145834 | 172433 | 201414 | 314225 | 521331 | 689470 | 901038 |
| 117021 | 146129 | 172534 | 201717 | 316633 | 522434 | 690152 | 901442 |
| 117728 | 146331 | 172635 | 201818 | 316835 | 523032 | 692964 | 902242 |
| 117930 | 146432 | 172938 | 202113 | 317332 | 524135 | 693764 | 905147 |
| 118023 | 146533 | 173132 | 202719 | 318637 | 525541 | 694362 | 907656 |
| 118124 | 146634 | 173334 | 203418 | 320826 | 529549 | 695768 | 908860 |
| 118225 | 146735 | 173435 | 203923 | 321323 | 529953 | 698168 | 910241 |
| 118528 | 146937 | 173536 | 204016 | 322224 | 531536 | 700634 | 910443 |
| 118831 | 147030 | 173637 | 204319 | 325129 | 536647 | 701535 | 911849 |
| 119025 | 147636 | 173738 | 204420 | 329844 | 540436 | 706040 | 912447 |
| 119126 | 147737 | 173839 | 204521 | 330324 | 541640 | 707749 | 917558 |
| 119732 | 148133 | 173940 | 204622 | 333330 | 545345 | 715041 | 919966 |
| 120414 | 148335 | 174841 | 205220 | 334635 | 547046 | 715950 | 922854 |
| 120515 | 148436 | 175136 | 206222 | 339847 | 548250 | 720337 | 923755 |
| 120717 | 148941 | 175237 | 206323 | 341834 | 549757 | 724446 | 926862 |
| 121315 | 149236 | 175338 | 206525 | 342129 | 550439 | 725751 | 927359 |
| 121416 | 149741 | 175540 | 206727 | 346137 | 552241 | 727553 | 929464 |
| 121618 | 149842 | 175742 | 206828 | 346945 | 553344 | 727654 | 930449 |
| 121921 | 150625 | 176037 | 206929 | 348545 | 555348 | 728454 | 933253 |
| 122317 | 150726 | 176441 | 207123 | 349244 | 555651 | 729254 | 942658 |

*Appendix 2—table 4 Continued on next page*

*Appendix 2—table 4 Continued*

| | | | | | | | |
|---|---|---|---|---|---|---|---|
| 122418 | 150928 | 176744 | 207426 | 350330 | 555954 | 731140 | 947668 |
| 122620 | 151021 | 176845 | 208024 | 352132 | 557857 | 734247 | 952863 |
| 122822 | 151324 | 177140 | 208125 | 352738 | 558657 | 735148 | 953764 |
| 123420 | 151425 | 177241 | 208327 | 353740 | 558960 | 737960 | 955465 |
| 123521 | 151728 | 177342 | 208428 | 355239 | 559457 | 742549 | 957974 |
| 123723 | 151829 | 177645 | 208630 | 356948 | 561444 | 744553 | 958976 |
| 123824 | 151930 | 178142 | 209127 | 358144 | 561949 | 748662 | 962058 |
| 123925 | 152225 | 178243 | 209228 | 360030 | 562345 | 749058 | 965771 |
| 124220 | 152427 | 178647 | 209329 | 361234 | 562446 | 751550 | 966975 |
| 124624 | 152831 | 178748 | 209531 | 361941 | 565452 | 753150 | 970764 |
| 124826 | 153025 | 178849 | 209834 | 362034 | 566454 | 757764 | 971160 |
| 125222 | 153126 | 178950 | 210011 | 365343 | 567052 | 759869 | 972566 |
| 125424 | 153227 | 179245 | 210112 | 366042 | 567961 | 760551 | 973770 |
| 126426 | 153631 | 179346 | 210415 | 368551 | 568963 | 763557 | 978578 |
| 126628 | 153732 | 179952 | 211114 | 368753 | 569965 | 765864 | 979984 |
| 127226 | 153833 | 180129 | 211215 | 376247 | 571144 | 766563 | 983773 |
| 127327 | 153934 | 180230 | 211316 | 377451 | 571548 | 769064 | 987074 |
| 127630 | 154229 | 180432 | 211619 | 378756 | 572045 | 770352 | 989987 |
| 127731 | 154330 | 180533 | 211821 | 378857 | 573249 | 771354 | 990366 |
| 127832 | 154532 | 180735 | 211922 | 379657 | 573451 | 773257 | 991267 |
| 128026 | 154734 | 180836 | 212015 | 380036 | 576255 | 774663 | 992673 |
| 128127 | 154835 | 180937 | 212116 | 381038 | 578057 | 779370 | 993675 |
| 128329 | 154936 | 181131 | 212217 | 381543 | 578158 | 782561 | 996782 |
| 128935 | 155231 | 181232 | 212419 | 382242 | 579867 | 783462 | 788674 |

HCP = Human Connectome Project.

**Appendix 2—table 5.** Subject IDs from the 209 PPMI controls with DaT SPECT data used in our analysis.

| PPMI subject ID | Image ID DaT SPECT | PPMI subject ID | Image ID DaT SPECT | PPMI subject ID | Image ID DaT SPECT |
|---|---|---|---|---|---|
| 3000 | 323662 | 3350 | 339901 | 3637 | 388521 |
| 3004 | 341194 | 3351 | 339902 | 3639 | 388523 |
| 3008 | 341195 | 3353 | 339904 | 3651 | 339008 |
| 3009 | 341196 | 3355 | 341236 | 3651 | 355956 |
| 3011 | 341198 | 3357 | 339907 | 3656 | 339014 |
| 3013 | 341200 | 3358 | 339908 | 3658 | 339016 |
| 3016 | 341202 | 3361 | 339911 | 3662 | 355221 |
| 3029 | 388468 | 3362 | 339912 | 3668 | 388528 |
| 3053 | 341207 | 3363 | 338780 | 3750 | 388535 |
| 3055 | 341209 | 3368 | 339917 | 3754 | 360616 |
| 3057 | 341211 | 3369 | 339918 | 3756 | 360617 |

*Appendix 2—table 5 Continued on next page*

*Appendix 2—table 5 Continued*

| PPMI subject ID | Image ID DaT SPECT | PPMI subject ID | Image ID DaT SPECT | PPMI subject ID | Image ID DaT SPECT |
|---|---|---|---|---|---|
| 3064 | 341217 | 3370 | 339919 | 3759 | 363950 |
| 3069 | 341221 | 3389 | 388504 | 3765 | 363951 |
| 3070 | 341222 | 3390 | 388505 | 3767 | 388536 |
| 3071 | 341223 | 3401 | 340345 | 3768 | 363952 |
| 3072 | 341224 | 3404 | 340346 | 3769 | 360618 |
| 3073 | 341225 | 3405 | 340347 | 3779 | 453700 |
| 3074 | 341226 | 3410 | 340351 | 3794 | 388545 |
| 3075 | 341227 | 3411 | 340352 | 3796 | 388147 |
| 3085 | 388470 | 3414 | 340354 | 3803 | 355230 |
| 3087 | 388472 | 3424 | 340363 | 3804 | 354344 |
| 3100 | 341230 | 3438 | 340388 | 3805 | 354345 |
| 3103 | 341233 | 3450 | 340398 | 3806 | 354346 |
| 3104 | 339536 | 3452 | 339923 | 3807 | 355231 |
| 3106 | 340418 | 3453 | 339924 | 3811 | 360620 |
| 3109 | 340423 | 3457 | 339928 | 3812 | 355232 |
| 3112 | 340426 | 3458 | 339929 | 3813 | 355233 |
| 3114 | 340430 | 3460 | 341243 | 3816 | 363953 |
| 3115 | 340431 | 3464 | 341245 | 3817 | 388148 |
| 3151 | 341018 | 3466 | 339932 | 3850 | 337832 |
| 3156 | 341021 | 3468 | 339934 | 3851 | 337833 |
| 3157 | 341022 | 3478 | 360613 | 3852 | 337834 |
| 3160 | 341023 | 3479 | 363945 | 3853 | 337835 |
| 3161 | 341024 | 3480 | 388509 | 3854 | 337836 |
| 3165 | 341027 | 3481 | 388510 | 3855 | 337445 |
| 3169 | 341031 | 3503 | 340400 | 3857 | 337837 |
| 3171 | 341033 | 3515 | 340408 | 3859 | 337839 |
| 3172 | 341034 | 3517 | 341248 | 3907 | 388556 |
| 3188 | 388483 | 3518 | 339537 | 3908 | 363957 |
| 3191 | 388486 | 3521 | 339539 | 3917 | 388563 |
| 3200 | 341036 | 3523 | 339541 | 3950 | 341083 |
| 3201 | 341037 | 3524 | 339542 | 3952 | 341085 |
| 3202 | 341038 | 3525 | 339543 | 3955 | 388565 |
| 3204 | 341040 | 3526 | 339544 | 3959 | 355241 |
| 3206 | 341042 | 3527 | 339545 | 3965 | 388573 |
| 3208 | 341044 | 3541 | 355215 | 3966 | 388574 |
| 3213 | 341049 | 3543 | 363946 | 3967 | 388576 |
| 3215 | 341051 | 3544 | 388514 | 3968 | 388577 |
| 3216 | 341052 | 3551 | 339550 | 3969 | 388578 |

*Appendix 2—table 5 Continued on next page*

*Appendix 2—table 5 Continued*

| PPMI subject ID | Image ID DaT SPECT | PPMI subject ID | Image ID DaT SPECT | PPMI subject ID | Image ID DaT SPECT |
|---|---|---|---|---|---|
| 3217 | 341053 | 3554 | 339552 | 4004 | 339032 |
| 3219 | 341055 | 3554 | 358138 | 4007 | 339035 |
| 3221 | 341057 | 3555 | 339553 | 4008 | 339036 |
| 3222 | 341058 | 3563 | 339559 | 4009 | 339037 |
| 3235 | 388488 | 3565 | 339561 | 4010 | 339038 |
| 3237 | 388490 | 3569 | 339564 | 4014 | 389268 |
| 3257 | 341067 | 3570 | 339565 | 4018 | 339045 |
| 3260 | 341068 | 3571 | 389245 | 4032 | 388583 |
| 3264 | 341070 | 3572 | 338781 | 4063 | 355246 |
| 3270 | 341074 | 3576 | 338785 | 4067 | 388593 |
| 3271 | 341075 | 3600 | 338788 | 4079 | 388596 |
| 3274 | 341077 | 3611 | 338797 | 4090 | 343886 |
| 3276 | 341079 | 3613 | 338799 | 4095 | 354353 |
| 3277 | 388491 | 3614 | 338800 | 4100 | 360623 |
| 3286 | 388494 | 3615 | 338801 | 4104 | 363963 |
| 3300 | 339889 | 3619 | 339001 | 4105 | 388600 |
| 3301 | 339890 | 3620 | 339002 | 4116 | 388613 |
| 3310 | 339896 | 3624 | 341251 | 4118 | 388615 |
| 3316 | 342187 | 3627 | 342204 | 4139 | 388627 |
| 3318 | 342189 | 3635 | 388519 | 4140 | 388628 |
| 3320 | 342191 | 3636 | 388520 | | |

PPMI = Parkinson's Progression Markers Initiative; DaT = dopamine transporter; SPECT = single-photon emission computed tomography.

**Appendix 2—table 6.** Subject IDs from PD patients and controls with resting-state fMRI data and DaT SPECT data from the PPMI dataset used in our analysis.

| PPMI subject ID | Image ID DaT SPECT | Image ID MRI | Diagnosis |
|---|---|---|---|
| 3310 | 339896 | 369414 | Control |
| 3318 | 342189 | 374882 | Control |
| 3350 | 339901 | 515208 | Control |
| 3351 | 339902 | 508245 | Control |
| 3353 | 339904 | 515216 | Control |
| 3361 | 339911 | 581042 | Control |
| 3369 | 339918 | 544617 | Control |
| 3389 | 388504 | 367349 | Control |
| 3551 | 339550 | 548987 | Control |
| 3563 | 339559 | 548989 | Control |
| 3565 | 339561 | 560369 | Control |
| 3769 | 360618 | 362609 | Control |

*Appendix 2—table 6 Continued on next page*

*Appendix 2—table 6 Continued*

| PPMI subject ID | Image ID DaT SPECT | Image ID MRI | Diagnosis |
| --- | --- | --- | --- |
| 4018 | 339045 | 365285 | Control |
| 4032 | 388583 | 367390 | Control |
| 3107 | 419849 | 378215 | PD |
| 3108 | 419850 | 378223 | PD |
| 3116 | 418649 | 366137 | PD |
| 3116 | 419854 | 417052 | PD |
| 3118 | 418470 | 362555 | PD |
| 3118 | 446107 | 430138 | PD |
| 3119 | 418650 | 382277 | PD |
| 3119 | 446108 | 430147 | PD |
| 3120 | 418651 | 374854 | PD |
| 3122 | 419241 | 382284 | PD |
| 3123 | 418652 | 382289 | PD |
| 3123 | 449008 | 440114 | PD |
| 3124 | 418653 | 387304 | PD |
| 3124 | 449009 | 440118 | PD |
| 3125 | 418654 | 387314 | PD |
| 3125 | 449010 | 440128 | PD |
| 3126 | 418655 | 397752 | PD |
| 3126 | 449011 | 440131 | PD |
| 3128 | 419553 | 395434 | PD |
| 3128 | 504427 | 466848 | PD |
| 3130 | 360608 | 355962 | PD |
| 3130 | 419554 | 417000 | PD |
| 3132 | 436066 | 423718 | PD |
| 3132 | 504428 | 498892 | PD |
| 3134 | 388480 | 369013 | PD |
| 3134 | 436067 | 436351 | PD |
| 3327 | 389212 | 362478 | PD |
| 3327 | 486550 | 412180 | PD |
| 3332 | 388500 | 378540 | PD |
| 3352 | 418905 | 372319 | PD |
| 3354 | 418906 | 372327 | PD |
| 3359 | 419866 | 397593 | PD |
| 3360 | 419867 | 393662 | PD |
| 3364 | 419868 | 393672 | PD |
| 3365 | 419659 | 397597 | PD |
| 3366 | 419869 | 397624 | PD |

*Appendix 2—table 6 Continued*

| PPMI subject ID | Image ID DaT SPECT | Image ID MRI | Diagnosis |
|---|---|---|---|
| 3367 | 419870 | 393674 | PD |
| 3371 | 418673 | 365166 | PD |
| 3372 | 436070 | 369487 | PD |
| 3372 | 446121 | 420330 | PD |
| 3373 | 418674 | 387316 | PD |
| 3373 | 449019 | 440174 | PD |
| 3374 | 418675 | 393614 | PD |
| 3374 | 446122 | 430165 | PD |
| 3377 | 418677 | 393628 | PD |
| 3377 | 449020 | 440186 | PD |
| 3378 | 418678 | 387324 | PD |
| 3380 | 418679 | 393636 | PD |
| 3380 | 468270 | 449575 | PD |
| 3383 | 355208 | 351070 | PD |
| 3383 | 419560 | 415707 | PD |
| 3385 | 360612 | 353398 | PD |
| 3385 | 436861 | 415713 | PD |
| 3386 | 388502 | 369048 | PD |
| 3387 | 389214 | 357590 | PD |
| 3387 | 436071 | 417033 | PD |
| 3392 | 388507 | 372995 | PD |
| 3392 | 442969 | 436390 | PD |
| 3552 | 418922 | 378354 | PD |
| 3556 | 418923 | 372348 | PD |
| 3556 | 504848 | 482323 | PD |
| 3557 | 504849 | 482329 | PD |
| 3559 | 418926 | 372359 | PD |
| 3559 | 504850 | 491605 | PD |
| 3574 | 419676 | 414623 | PD |
| 3575 | 419677 | 581115 | PD |
| 3575 | 418690 | 365225 | PD |
| 3585 | 449026 | 440198 | PD |
| 3586 | 468275 | 449581 | PD |
| 3587 | 468276 | 449584 | PD |
| 3591 | 388516 | 373018 | PD |
| 3591 | 504435 | 491626 | PD |
| 3592 | 388517 | 373035 | PD |
| 3592 | 442973 | 436404 | PD |

*Appendix 2—table 6 Continued*

| PPMI subject ID | Image ID DaT SPECT | Image ID MRI | Diagnosis |
| --- | --- | --- | --- |
| 3593 | 388518 | 369096 | PD |
| 3593 | 436073 | 430199 | PD |
| 3593 | 504436 | 507400 | PD |
| 3758 | 418698 | 374893 | PD |
| 3758 | 419880 | 402067 | PD |
| 3760 | 418499 | 362591 | PD |
| 3787 | 419576 | 412194 | PD |
| 3800 | 389258 | 393684 | PD |
| 3808 | 419885 | 402071 | PD |
| 3815 | 419886 | 581145 | PD |
| 3818 | 446139 | 440242 | PD |
| 3819 | 419270 | 395448 | PD |
| 3822 | 419271 | 382366 | PD |
| 3822 | 449035 | 440262 | PD |
| 3823 | 419272 | 395585 | PD |
| 3823 | 449036 | 440267 | PD |
| 3824 | 419579 | 395592 | PD |
| 3824 | 468279 | 449614 | PD |
| 3825 | 419273 | 393639 | PD |
| 3825 | 504450 | 549048 | PD |
| 3826 | 419274 | 395600 | PD |
| 3826 | 468280 | 449625 | PD |
| 3828 | 419580 | 395605 | PD |
| 3828 | 468281 | 449661 | PD |
| 3829 | 419581 | 395614 | PD |
| 3830 | 419582 | 412202 | PD |
| 3830 | 495006 | 468929 | PD |
| 3831 | 419583 | 402267 | PD |
| 3832 | 419584 | 412209 | PD |
| 3832 | 495007 | 468935 | PD |
| 3834 | 419585 | 415724 | PD |
| 3834 | 504454 | 473094 | PD |
| 3835 | 436875 | 415731 | PD |
| 3838 | 436075 | 423748 | PD |
| 3838 | 504456 | 515249 | PD |
| 3869 | 436077 | 415744 | PD |
| 3870 | 363956 | 395313 | PD |
| 3870 | 486557 | 415751 | PD |

*Appendix 2—table 6 Continued on next page*

*Appendix 2—table 6 Continued*

| PPMI subject ID | Image ID DaT SPECT | Image ID MRI | Diagnosis |
|---|---|---|---|
| 4005 | 419890 | 397646 | PD |
| 4011 | 418504 | 402285 | PD |
| 4019 | 418710 | 362640 | PD |
| 4019 | 446143 | 417057 | PD |
| 4021 | 419277 | 430178 | PD |
| 4022 | 418712 | 365294 | PD |
| 4022 | 446145 | 417065 | PD |
| 4029 | 468288 | 468943 | PD |
| 4030 | 363959 | 356036 | PD |
| 4030 | 419596 | 415756 | PD |
| 4030 | 495322 | 468949 | PD |
| 4034 | 388585 | 367425 | PD |
| 4034 | 436083 | 423755 | PD |
| 4035 | 388587 | 369183 | PD |
| 4035 | 436084 | 423762 | PD |
| 4035 | 504466 | 475680 | PD |
| 4038 | 388590 | 367446 | PD |
| 4038 | 436085 | 430210 | PD |

PD = Parkinson's disease; DaT = dopamine transporter; fMRI = functional MRI; SPECT = single-photon emission computed tomography; PPMI = Parkinson's Progression Markers Initiative.

