## [Editor Report]

The authors make a convincing argument that they have found an MRI-based biomarker for dopaminergic input into the striatum. Because the dopaminergic system is involved in neurodegenerative disorders such as Parkinson's disease and also in processing reward signals, the biomarker is likely to become widely adopted and enable new types of experiments in related fields. In this revision, the authors further demonstrate the specificity of the potential biomarker and its lack of sensitivity to head motion.

---

## [Decision Letter]

**Decision letter after peer review:**

Thank you for submitting your article "Mapping dopaminergic projections in the human brain with resting-state fMRI" for consideration by *eLife*. Your article has been reviewed by 3 peer reviewers, including Shella Keilholz as Reviewing Editor and Reviewer #1, and the evaluation has been overseen by Michael Frank as the Senior Editor. The following individuals involved in review of your submission have agreed to reveal their identity: Georg Northoff MD, PhD (Reviewer #2); Finnegan J Calabro (Reviewer #3).

Essential revisions:

1) A key assertion of the manuscript is that the biomarker is specific to the dopaminergic system. This assertion needs stronger support. Studies that show that the biomarker disambiguates between different neuromodulatory systems (e.g., serotonergic) are needed to address this issue. For example, existing data could be examined by putting the seed in the subcortical regions like Raphe nucleus and VTA/SN and then investigating their upstream functional connectivity (see Martino et al., 2020, Conio et al., 2020, and others). The results could be compared to the present DA-driven results including conjunction and exclusive masking.

2) Common confounds for resting state fMRI analysis (e.g., head motion) need to be better addressed in terms of how they affect the mode that serves as a biomarker. It is essential to be certain that the biomarker is not picking up motion differences between groups.

*Reviewer #1 (Recommendations for the authors):*

It was not clear to me how the modes for different parts of the striatum were combined. A brief explanation would be useful.

Have the authors looked at functional connectivity between areas that exhibit differences in the 2^nd^ mode and their hypothesized targets? For example, if the area that exhibits differences as a function of alcohol use also show differences in functional connectivity to a target area that replicates previous studies, it would further strengthen the manuscript.

It would be informative to see if other modes are altered in PD. If they are not, it would suggest great specificity for the 2^nd^ mode.

There’s an interesting difference between the overall correlation of DaT and the 2^nd^ order mode in the putamen as compared to the caudate that should be discussed.

Any overlap between the tobacco and alcohol use group should be described.

*Reviewer #2 (Recommendations for the authors):*

– The connectopic mapping is based on functional connectivity and correlating time series. May be analysis of dynamic functional connectivity could enhance the validity of the data: if cortical regions show similar dynamic pattern int heir variability, it could be used to further specify the specificity of the cortical connectivity pattern of the striatum.

– May be a figure of the differential cortical connectivity patterns of the three modes (zero-first-, and second-order) (Figure 1) could be shown as that would reveal the cortical specificity of the striatal subregions which is biochemically relevant…

– As I understand the DaT SPECT subjects are different from the HCP subjects so you correlate different healthy subjects with each other…correct? If so, I would recommend at least some healthy subjects to have both SPECT and fMRI (beyond the PD subjects)…..even if a low number, it would increase the validity of the marker….this is important given that, as far as I can see in the tables, there is considerable inter-subject variability in the striatal data both SPECT and fMRI.

– Figure 3: where are the cortical connections in the three fMRI striatal modes? Do they correlate with the DaT SPECT striatal data?

– Also: the HCP sequences and scanning measures are not ideal to capture subcortical regions like the striatum including their subregions as they do not, as far as I recall, contain axial slices….. this could be mentioned as limitation…

– Statistically: the main results on this paper rely on correlation mostly Pearson. It would be nice to have that further solidified by using more robust regression analyses….

*Reviewer #3 (Recommendations for the authors):*

It is unclear to me how the topographic maps shown in Figure 1 are derived, and specifically how these relate to the spatial fits being performed separately for each region. I would have expected discontinuities in these maps at the regional boundaries, but the maps appear to vary continuously across the striatum, so some clarification on what these maps are representing relative to the per-ROI statistical surface odelling being performed would be helpful.

The resulting second order mode seems qualitatively similar to maps found in previous connectivity mapping approaches (e.g., Tziortzi et al., 2014). Some discussion about either consistency with previous approaches, or description of differences in what this method identifies, would be helpful. In particular, if the patterns are sufficiently similar, this would open the possibility of associating analyses performed using these other atlases for interpretations related to DA distribution. If there are differences, it would be interesting to discuss how these methods differ in the patterns they identify.

I’m confused by the description of a “lossless” SVD for dimensionality reduction (l 543). Presumably to attain a reduction in the matrix size, some proportion of eigenvalues are retained with the rest removed, rendering it lossy. Some clarification here, or information about what proportion is retained, would be helpful.

What is the rationale for using a scree test to choose the best TSM, rather than a less subjective AIC/BIC or similar? The selection of such complex models, in relatively small spatial regions (e.g., quartic model for caudate-Nacc) raises questions about how effectively extra coefficients are being penalized in this approach.

A citation is needed for this assertion: “as PD is known to affect the putamen region of the striatum before the caudate-Nacc region” (l. 215)

---

## [Author Response]

Essential revisions:1) A key assertion of the manuscript is that the biomarker is specific to the dopaminergic system. This assertion needs stronger support. Studies that show that the biomarker disambiguates between different neuromodulatory systems (e.g., serotonergic) are needed to address this issue. For example, existing data could be examined by putting the seed in the subcortical regions like Raphe nucleus and VTA/SN and then investigating their upstream functional connectivity (see Martino et al., 2020, Conio et al., 2020, and others). The results could be compared to the present DA-driven results including conjunction and exclusive masking.

We thank the reviewers and editor for this suggestion and agree that providing additional support for the specificity of our potential biomarker to the dopaminergic system would strengthen our results. However, the “typical” seed-based connectivity analyses that are proposed reflect the “average” functional connectivity of one region with another region in the brain, and by using such a seed-based approach there may still multiple (neurotransmitter) systems involved due to mesoscopic nature of connectivity within each relatively large voxel. As such, we decided to directly look at the level of neuromodulatory systems by investigating the neuroreceptor/neurotransporter architecture obtained with PET. More specifically, to be able to demonstrate that the association of the second-order striatal connectivity mode with the DaT SPECT scan is stronger than that of any other neurotransmitter system, we investigated the spatial correspondence of the second-order connectivity mode to a large set of PET markers indexing various neuromodulatory systems. To this end, we made use of various PET scans tapping into different neurotransmitter systems (group-averages of 11-36 controls) obtained from the publicly available JuSpace toolbox (Dukart et al., 2021; https://github.com/juryxy/ JuSpace). The included PET scans for the serotonin system are: 5HT1a receptor (5HT1a_WAY), 5HT1b receptor (5HT1b_P943), 5HT2a receptor (5HT2a_ALT), and the serotonin transporter (SERT_DASB and SERT_MADAM). The included PET scans for the dopamine system (other than DaT SPECT) are: Dopamine type 1 receptor (D1_SCH23390), Dopamine type 2 receptor (D2_RACLOPRIDE), and FDOPA (FDOPA_f18). In addition, the following neurotransmitter systems were also included: for GABA the GABAA receptor (GABAA_FLUMAZENIL), for Noradrenalin the Noradrenalin receptor (NAT_MRB) and for the opiod system the mu opiod receptor (MU_CARFENTANIL).

We applied trend surface modelling (TSM) to each of these PET scans in the striatum (as in the original submission the rest of the brain was masked and not included) and computed correlations between TSM coefficients modelling the second-order connectivity mode and the TSM coefficients modelling these PET-derived markers. For each PET scan, the correlations with the TSM coefficients were normalized using the Fisher r-to-z transformation and the absolute correlation was taken. These normalized correlations are visualized in Figure 2—figure supplement 1 of the revised manuscript. As can be observed, the correlation of the second-order connectivity mode with the DaT SPECT scan is substantially higher than that of any other PET marker. Though it is noticeable that some of the correlations with the PET markers for the serotonin system (SERT_DASB transporter and 5HT1b receptor) that are also known to have a high density in the striatum, are also relatively high. Nevertheless these markers only reach about half of the correlation value of DaT SPECT. To support the robustness and significance of the correlation of our second-order connectivity mode with the DaT SPECT scan *over and above* the correlation with the markers of the serotonergic system, we tested the correlation between the TSM coefficients obtained for the second-order connectivity mode and the DaT SPECT scan in striatum for significance using permutation testing (*N=10000*). More specifically, we permuted corresponding TSM coefficients obtained for each of the PET markers and thereby generated a null distribution by computing the absolute (Fisher r-to-z normalized) correlations between the connectivity mode TSM coefficients and the permuted TSM coefficients of the other PET markers. Permutations were conducted separately for each coefficient, not permuting across coefficients to ensure interchangability under the null assumption of no differentiation across different PET markers.

As can be observed in the bottom figure, all permuted correlations are lower than the correlation observed between the DaT SPECT scan and the connectivity mode, indicating that the observed correlation between the connectivity mode and DaT SPECT scan is highly significant and unlikely to be obtained by chance. Furthermore, using this null distribution, we defined the Bonferoni-corrected threshold for significance corresponding to *p*=0.0008 (i.e., *p*=0.01/12 PET and SPECT scans), which we added to the top figure displaying the correlations with the other PET tracers from the Juspace toolbox. Of note, the correlation with the dopaminergic D1 and D2 receptors, as opposed to DaT SPECT is not particularly high. However, this is not surprising since these receptors are present on postsynaptic neurons and are likely representative of postsynaptic dopaminergic projections from striatum to cortex, rather than the presynaptic dopaminergic projections from the midbrain to the striatum reflected by the DaT SPECT scan.

We have included the above description and figure (Figure 2—figure supplement 1) in the Supplementary Materials. In addition, we have also added the following sections to the results and Discussion sections of our manuscript:

Results, page 6 (lines 163-168):

“Finally, to further demonstrate the high specificity of the second-order connectivity mode to the DaT SPECT scan, we computed correlations with the TSM coefficients of all PET scans, tapping into various neurotransmitter systems, included in the publicly available JuSpace toolbox (Dukart et al., 2021). Figure 2—figure supplement 1 reveals that the correlation between the TSM coefficients of the second-order connectivity mode with the DaT SPECT scan is not only highly significant but also significantly higher than the correlations with the TSM coefficients of any other PET scan.”

Discussion, pages 12 (lines 353-359):

“We further demonstrated that the association of the second-order striatal connectivity mode with the DaT SPECT scan is stronger than that of all the other investigated PET markers indexing various neurotransmitter systems, see Figure 2—figure supplement 1. This figure also shows that correlations of the second-order connectivity mode with dopamine receptors D1 and D2 in striatum, which are present on postsynaptic dopaminergic neurons, were substantially lower and not significant (r=-0.290, p=0.086 and r=0.241, p=0.156), suggesting that the second-order connectivity mode is specific to presynaptic dopaminergic projections.”

2) Common confounds for resting state fMRI analysis (e.g., head motion) need to be better addressed in terms of how they affect the mode that serves as a biomarker. It is essential to be certain that the biomarker is not picking up motion differences between groups.

We agree with the reviewers that head motion can have a profound impact on resting-state functional connectivity analyses if not properly controlled for. However, we would first like to point out that the amount of head motion in all the samples we investigated was rather low to start with and very comparable between the different samples, see Author response table 1. Furthermore, the resting-state data of the HCP dataset was thoroughly corrected for head-motion related artifacts by applying ICA-FIX (Salimi-Khorshidi et al., 2014). Similarly, in the PPMI dataset and in our local PD dataset we thoroughly correct for head motion-related artifacts using ICA-AROMA (Pruim et al., 2015), which has been demonstrated to remove head motion-related artifacts with high accuracy while preserving signal of interest (Parkes et al., 2017). In our local PD dataset we moreover removed any residual head-motion artefacts from the data by regressing out the EMG parameters used for monitoring potential tremor-related activity. Nevertheless and in order to rule out that our findings are related to residual head motion, we conducted two additional post-hoc sensitivity analyses, which are also described below, both for the group level findings and the within-subject associations.

**Author response table 1. sa2table1:** 

		meanFD		
	*N*	*Mean*	*SD*	*Range*
**HCP**	839	0.0871	0.0362	0.0375-0.3155
**PPMI all DaT SPECT controls**	209	*Not available for DaT SPECT scan*		
**PPMI within-subject analysis, full sample**	144	0.1308	0.0620	0.0438-0.3124
**PPMI within-subject analysis, patients**	130	0.1292	0.0609	0.0438-0.3124
**PPMI within-subject analysis, controls**	14	0.1463	0.0541	0.0722-0.2778
**Local PD dataset, full sample placebo**	59	0.1104	0.0535	0.0352-0.2762
**Local PD dataset, patients placebo**	39	0.1165	0.0492	0.0490-0.2363
**Local PD dataset, controls placebo**	20	0.0987	0.0604	0.0352-0.2762
**Local PD dataset, patients LDOPA**	39	0.1277	0.0825	0.0376-0.5252*
*There was 1 subject with a meanFD of 0.5252, but also after excluding this subject results remained significant. All other subjects had a meanFD<0.3.				

Sensitivity analysis: group-level mapping connectivity mode onto DaT SPECT

To demonstrate that the mapping of the group-average second order connectivity mode onto the group-average DaT SPECT scan (*r*=0.884) was not influenced by residual head motion, we generated connectivity modes for the 10% highest movers (N=84, meanFD range: 0.0376-0.0538) and 10% lowest movers (N=84, meanFD range: 0.1354-0.3155) of the HCP dataset and computed the spatial correlation to the group-average DaT SPECT scan (N=209, no head motion metrics available, so we used the entire sample). This analysis revealed very similar connectivity modes, see Figure 2—figure supplement 2 of the revised manuscript and a very similar spatial correlation for the low FD group (*r*=0.883) and the high FD group (*r*=0.886), indicating that this mapping was not induced by residual head motion.

Sensitivity analysis: within-subject mapping connectivity mode and DaT SPECT

We divided the subsample of the PPMI dataset where both a resting-state fMRI scan and DaT SPECT (used for mapping the second-order connectivity mode onto the DaT SPECT at the within-subject level) is available in half based on the median of the meanFD (*meanFD = 0.126*) and computed the within-subject correlation between the DaT SPECT and connectivity mode for the ‘low motion’ and ‘high motion’ halfs of the sample separately. This analysis revealed that the within-subject spatial correlation between the DaT SPECT and second-order connectivity mode was virtually identical between the relatively low and relatively high meanFD sample, now included as Figure 3—figure supplement 2 of the revised manuscript. It does appear that for the caudate-NAcc region the correlations between the connectivity mode and the DaT SPECT scan are slightly lower for the relatively high motion group than for the relatively low motion group. This indicates that the high spatial correlation of the connectivity mode with the DaT SPECT scan is not artificially induced by head motion, but that residual head motion instead weakens this correlation (i.e. it ads noise leading to an underestimation of the true spatial correspondence). All these additional analyses thus suggest that the proposed biomarker is not picking up on residual motion.

These additional analyses are now described in the “Post-hoc analyses of residual head motion” paragraph of appendix 1 and the results are shown in Figure 2—figure supplement 2 and Figure 3—figure supplement 2.

Reviewer #1 (Recommendations for the authors):It was not clear to me how the modes for different parts of the striatum were combined. A brief explanation would be useful.

We apologize for the opacity. All analyses in this paper were conducted for each striatal ROI separately, but in Figures 1 and 2 the four striatal modes have been combined purely for visualization purposes. This was achieved by loading the striatal modes simultaneously in FslView from which the figures were derived. This was possible because the four striatal ROIs did not spatially overlap and were each normalized to have values (indicating variations in connectivity profiles) ranging from 0 to 1.

We have adjusted the following sections on pages 4 and 5 of the manuscript to clarify this procedure:

Lines 118-120: “For all analyses described in this paper, connectopic mapping was applied to the left and right putamen and caudate-NAcc subregions separately to increase regional specificity and the second-order striatal connectivity mode was selected for each of the four striatal ROIs.”

Lines 140-141: “The modes for left and right putamen and caudate-NAcc have been combined in this figure (i.e., the four ROIs were loaded in FslView simultaneously from which the below figures were derived) to aid …”

Have the authors looked at functional connectivity between areas that exhibit differences in the 2nd mode and their hypothesized targets? For example, if the area that exhibits differences as a function of alcohol use also show differences in functional connectivity to a target area that replicates previous studies, it would further strengthen the manuscript.

In this paper, we hypothesized that the mapping between the second-order connectivity mode and the DaT SPECT scan reflects dopaminergic projections from the midbrain (SN and VTA) to the striatum. In theory we could investigate connectivity between the midbrain and striatum. However, we do not know how our connectivity mode relates to cortical connectivity, which we assume the reviewer is referring to. Due to methodological constraints, it is unfortunately extremely difficult, if not impossible, to obtain and investigate functional connectivity specific to the second-order mode. That is, functional connectivity between two regions represents the synchronicity between the ‘average’ BOLD signal in region A and the ‘average’ BOLD signal in region B. One key feature of our novel analysis approach, however, is that our connectivity modes are ‘overlapping’ in the striatum. While this addresses the functional multiplicity across ROIs it suffers from the fact that the only way to obtain the cortical connectivity patterns, i.e., targets, specific to one particular connectivity mode would be to do a regression analysis in which we correct for the other connectivity modes. This has previously been shown to not be useful because a lot of signal is removed/regressed out. Accordingly, it is difficult to empirically investigate the unique (cortical) projections of only the second-order connectivity mode. The approach proposed is further limited by the fact that connectivity modes are defined to index a gradient, reflecting a gradual change in the connectivity pattern. These gradients by definition are not associated with a single projection but instead reflect a spectrum of projections. Such a view on functional connectivity is very unique and we therefore cannot think of any existing study using more classical (non-overlapping, non-gradual) characterizations that we can replicate in this fashion. For more information please see our 2020 review paper in Neuroimage on interpreting functional connectivity results in the face of functional heterogeneity and multiplicity (Haak and Beckmann 2020).

It would be informative to see if other modes are altered in PD. If they are not, it would suggest great specificity for the 2nd mode.

We have previously related the first-order striatal connectivity mode to cortico-striatal projections (Marquand et al., 2017) and prior work in PD has consistently shown that cortico-striatal connectivity is altered in PD (for reviews see Filippi et al., 2019, Movement disorders; Tessitore et al., 2019, Journal of Parkinson’s Disease). Such studies have found that connectivity of the (posterior) putamen with (motor) cortex is decreased in PD patients compared to controls, which is thought to be secondary effect related to the degeneration of dopaminergic neurons in the putamen. Consistent with these findings, when we conduct our analyses for the first-order connectivity mode this also revealed a significant difference for the putamen region between the control group and right-dominant PD group (*Χ2*=33.342, *p*=0.015), as well as a significant difference between the control group and left-dominant PD group (*Χ2*=54.040, *p*<0.001). Since the first-order connectivity maps onto cortico-striatal connectivity, alterations in connectivity of putamen are likely reflected in this connectivity mode. While an absence of a significant difference for the first-order connectivity mode of course would have suggested greater specificity for the second order mode, the fact that the first-order mode is also different in PD, is not surprising.

There's an interesting difference between the overall correlation of DaT and the 2nd order mode in the putamen as compared to the caudate that should be discussed.

While the mean within-subject correlation of DaT and the second-order mode across the four striatal subregions is 0.58, the reviewer has correctly noticed that this correlation is higher in left and right putamen (r=0.61/0.62) than in the left and right caudate region (0.51/0.44; see Figure 3). We speculate that this difference might relate to three potential factors:

1) The putamen contains higher levels of dopamine than the caudate-NAcc (Hortnagl et al., 2020) which could be the result of more dopaminergic projections to the putamen and the putamen might therefore also be more strongly associated with dopamine-related signaling than the caudate-NAcc.

2) The putamen has more voxels (N=2171) than caudate-NAcc (N=1678) (in MNI152 2mm space) and therefore the resting-state fMRI signal in putamen is likely more stable than signal in caudate-NAcc.

3) The caudate-NAcc lies directly next to the ventricles, resulting in a high-intensity border (CSF vs gray matter) on the MRI scan, whereas putamen lies a bit further away from the ventricles. As such, (small amounts of) head motion of the participant will disturb the signal of the caudate-NAcc more than for putamen.

We have added the following section to the discussion (page 13, lines 380-386) of the manuscript:

“With respect to Figure 3, we further note the difference in the within-subject correlation for the putamen (r=0.61/0.62) compared to caudate-NAcc region (r=0.51/0.44). We tentatively speculate that this difference might relate to a stronger and more stable dopamine-related resting-state fMRI signal in putamen compared to caudate-NAcc resulting from more dopaminergic projections to putamen (Hortnagl et al., 2020), and the putamen being larger in size and spatially further away from the ventricles and therefore less susceptible to motion-related artefact than the caudate-NAcc region.”

Any overlap between the tobacco and alcohol use group should be described.

A total of 5 subjects were included in both the tobacco use and alcohol use analyses. We post-hoc excluded these participants and repeated both analyses, which showed that both the association with alcohol use (*Χ2*=39.40, *p*=0.025) and with tobacco use (*Χ2*=62.01, *p*<0.001) were still significant. This means that our results were stable and thus not driven by the excluded, overlapping participants. We have added the following sentence to the manuscript (page 10, lines 308-310):

“Finally, five subjects were included in both the tobacco and alcohol use analyses but the associations with tobacco use (Χ2=39.40, p=0.025) and alcohol use (Χ2=62.01, p<0.001) also remained significant after excluding these subjects.”

Reviewer #2 (Recommendations for the authors):– The connectopic mapping is based on functional connectivity and correlating time series. May be analysis of dynamic functional connectivity could enhance the validity of the data: if cortical regions show similar dynamic pattern int heir variability, it could be used to further specify the specificity of the cortical connectivity pattern of the striatum.

We agree with the reviewer that an analysis of dynamical functional connectivity of the striatum is very interesting and indeed we are currently adapting the connectopic mapping methodology to look into dynamically changing connectivity gradients. This work remains to be validated and therefore we believe such an analysis would be beyond the scope of this paper. Given that none of the data sets used in our current manuscript features an intervention during the course of the data acquisition, however, we belief that our future methodological advances (and indeed even other types of ‘dynamic functional connectivity’) would be hard to interpret, let alone validate. As such, for this data these approaches are unlikely to enhance the validity of our results since the striatal connectivity modes are defined based on their functional connectivity profile with the rest of the brain, including the cortex.

– May be a figure of the differential cortical connectivity patterns of the three modes (zero-first-, and second-order) (Figure 1) could be shown as that would reveal the cortical specificity of the striatal subregions which is biochemically relevant…

We recognize that showing the differential connectivity patterns of the three different striatal connectivity modes would be very informative. Yet, as already mentioned in response to the second comment of reviewer 1, due to methodological constraints it is unfortunately extremely difficult, if not impossible, to obtain and visualize these connectivity patterns for each specific striatal connectivity mode. That is, functional connectivity between two regions represents the synchronicity between the ‘average’ BOLD signal in region A and the ‘average’ BOLD signal in region B. One key feature of our novel analysis approach, however, is that our connectivity modes are ‘overlapping’ in the striatum. While this addresses the functional multiplicity across ROIs, it suffers from the fact that the only way to obtain the cortical connectivity patterns specific to one particular connectivity mode would be to do a regression analysis in which we correct for the other connectivity modes. Indeed, we conducted such analyses for striatum and anterior temporal lobe in Marquand et al., 2017 and a later preprint (https://www.biorxiv.org/content/10.1101/2020.05.28.121137v1) and showed that while this regression approach works reasonable well for obtaining the cortical projections for the primary, dominant gradient, it leads to very noisy estimates for higher order gradients, where the dominant gradient needs to be regressed. Accordingly, it is difficult to empirically investigate the unique (cortical) projections of each specific connectivity mode. The approach proposed is further limited by the fact that connectivity modes are defined to index a gradient, reflecting a gradual change in the connectivity pattern. These gradients by definition are not associated with a single projection but instead reflect a spectrum of projections.

Therefore, rather than rely on cortical projections to address the specificity of the findings, in our response to comment 1 in the essential revisions section we actually demonstrated biochemical specificity of the second-order mode to the DaT SPECT scan. We investigated the spatial correspondence of the second-order connectivity mode with multiple PET markers indexing various neuromodulatory systems and showed that the mapping of the second-order connectivity mode to the DaT SPECT scan was not only highly significant but also superior to all investigated markers (e.g., neurotransmitter-specific receptors and transporters) of other neurotransmitter systems.

– As I understand the DaT SPECT subjects are different from the HCP subjects so you correlate different healthy subjects with each other…correct? If so, I would recommend at least some healthy subjects to have both SPECT and fMRI (beyond the PD subjects)…..even if a low number, it would increase the validity of the marker….this is important given that, as far as I can see in the tables, there is considerable inter-subject variability in the striatal data both SPECT and fMRI.

We agree with the reviewer. Next to our initial analysis in which we mapped the group-level second-order connectivity mode obtained in the HCP dataset onto the group-level DaT SPECT obtained in the PPMI dataset, we demonstrate validity of this mapping at the *within-subject* level in a subsample of PPMI participants (130 datasets from PD patients and 14 from controls) with both a DaT SPECT scan and resting-state fMRI data available (see Figure 3 in the manuscript).

We apologise if it was not clear that these analyses were indeed based on subjects where both SPECT and fMRI data were available – we have now clarified this further in the figure caption. To summarize, within a smaller sample of PD patients and controls with good quality connectivity modes, we not only replicated the spatial correspondence between the connectivity mode and DaT SPECT scan at the group-level in both the PD group (*r*=0.714) and in the control group (*r*=0.721) but also observed a *within-subject* spatial correlation of 0.58 across the four striatal subregions (0.44>*r*<0.62; mean=0.58, 95% CI = [0.56,0.60]), see Figure 3. This latter analysis included both control (red dots) and PD subjects (black dots), and as can be observed Figure, correlations appear similar in the control group (despite being a very small group) and PD group.

– Figure 3: where are the cortical connections in the three fMRI striatal modes? Do they correlate with the DaT SPECT striatal data?

Figure 3 shows the spatial correlation between the second-order connectivity mode and the DaT SPECT scan for four striatal subregions across a subsample of PPMI controls and patients that had both a resting-state fMRI scan and DaT SPECT scan available. We have clarified this further in the figure caption.

– Also: the HCP sequences and scanning measures are not ideal to capture subcortical regions like the striatum including their subregions as they do not, as far as I recall, contain axial slices….. this could be mentioned as limitation…

The reviewer is correct that the scanning sequences used in the HCP dataset are multiband sequences that have been optimized to acquire signal in cortex and therefore are not ideal for investigating signals within subcortical structures. The reduced temporal signal-to-noise ratio in subcortex, however, has been demonstrated not to matter too much when characterizing average connectivity due to the extreme high N and otherwise exquisite data quality (Smith et al., 2013, Neuroimage: Resting-state fMRI in the Human Connectome Project). Indeed, the fact that we find multiple strong associations with connectivity modes in striatum implies that the signal-to-noise ratio is subcortical areas was still sufficiently high to detect these associations, and furthermore this also means that we likely underestimated the strength of these associations, given that low tSNR will predominantly impact on the sensitivity of the analysis.

We do now mention this as a limitation on page 14, lines 426-428 in the discussion of the manuscript:

“However, a limitation that should be mentioned is that the resting-state fMRI sequence of the HCP dataset has not been optimized for subcortical brain regions.”

– Statistically: the main results on this paper rely on correlation mostly Pearson. It would be nice to have that further solidified by using more robust regression analyses….

Indeed, in order to gain confidence over and above Pearson correlations we already tested all associations with behavior robustly using multiple linear regression analyses that included all the TSM coefficients modeling the second-order connectivity mode (we also refer to these analyses as omnibus tests in the manuscript, as for example in the second paragraph on page 7). In case such a regression analysis revealed a significant association, Pearson correlations with the individual TSM were computed post-hoc to visualize and interpret these relationships.

Reviewer #3 (Recommendations for the authors):It is unclear to me how the topographic maps shown in Figure 1 are derived, and specifically how these relate to the spatial fits being performed separately for each region. I would have expected discontinuities in these maps at the regional boundaries, but the maps appear to vary continuously across the striatum, so some clarification on what these maps are representing relative to the per-ROI statistical surface modeling being performed would be helpful.

To clarify, the connectivity modes shown in Figure 1 are the same as the modes shown in Figure 2. We indeed applied connectopic mapping to the putamen and caudate-NAcc regions separately to be able to investigate the second-order connectivity mode in these striatal subregions separately. In Marquand et al., 2017, we showed that the zero-order connectivity mode represents the anatomical subdivision of the striatum into putamen, caudate and NAcc. By focusing on the second-order gradient, we thus discount the variance associated with individual subnuclei and therefore we don’t see these strong discontinuities (i.e., the zero-order mode) anymore in the second-order mode, nor when investigating the putamen and caudate-NAcc separately and also not when combining these modes into one figure for visualization.

The resulting second order mode seems qualitatively similar to maps found in previous connectivity mapping approaches (e.g., Tziortzi et al., 2014). Some discussion about either consistency with previous approaches, or description of differences in what this method identifies, would be helpful. In particular, if the patterns are sufficiently similar, this would open the possibility of associating analyses performed using these other atlases for interpretations related to DA distribution. If there are differences, it would be interesting to discuss how these methods differ in the patterns they identify.

To the best of our knowledge there are currently no papers published that obtained and investigated higher-order (functional) connectivity gradients in striatum that can be compared to the second-order gradient obtained in our study. Several papers applying parcellation-based approaches to striatum have been published, yet these do not allow for overlapping (connectivity) modes but are based on the mean, dominant signal in striatum. We believe that to be able to obtain neurotransmitter specific distributions in the brain from structural or functional MRI data, the applied method needs to allow for overlapping connectivity profiles.

Tziortzi et al., (2014) generated a parcellation of the striatum based on structural connectivity with cortical systems derived from diffusion-weighted imaging data. This parcellation does indeed look somewhat similar to our connectivity mode. However, the figure displaying this parcellation (Figure 2) only consists of coronal views of striatum, making it difficult to define the exact spatial correspondence to our second-order striatal connectivity mode. Indeed, when comparing Figure 2 from the Tziortzi paper with the coronal slices for zeroth-order, first-order and second-order mode shown in Figure 2 (middle row, y=6) of our paper, it is difficult to conclude with which of these modes the Tziortzi parcellations show the highest spatial similarity, so we refrain from making comparisons with the Tziortzi paper in our article.

I'm confused by the description of a "lossless" SVD for dimensionality reduction (l 543). Presumably to attain a reduction in the matrix size, some proportion of eigenvalues are retained with the rest removed, rendering it lossy. Some clarification here, or information about what proportion is retained, would be helpful.

The term *lossless* in the context of an SVD is used to indicate that an SVD is applied where initially all eigenvalues and thus, all data, are being retained. Please note that within our framework we are not using a SVD for dimensionality reduction, but only for defining an orthonormal basis to ease computation.

What is the rationale for using a scree test to choose the best TSM, rather than a less subjective AIC/BIC or similar? The selection of such complex models, in relatively small spatial regions (e.g., quartic model for caudate-NAcc) raises questions about how effectively extra coefficients are being penalized in this approach.

The selection of the most appropriate model order is indeed a bit tricky in this case because of the high smoothness of the connectivity mode, which is inherent to the connectopic mapping procedure enforcing a gradient-like structure. This high smoothness in turn makes it difficult to estimate correct spatial degrees of freedom (DoF), which methods like AIC/BIC use for penalization. We actually have looked at AIC and BIC, but due to the smoothness of the connectivity mode they always picked the highest model order, meaning that these methods do not correctly penalize. Therefore we here used the scree test, which more effectively penalizes in this type of data. In addition, we repeated our analysis using model order 3 (that is a cubic model with 9 TSM coefficients) for both the putamen and caudate-NAcc regions, which gave us similar results: the correlation between the TSM coefficients modeling the second-order connectivity mode and the DaT SPECT scan is *r*=0.90, p<0.0001, which is comparable to our original finding (*r*=0.925, p<0.0001). This thus indicates that our findings do not heavily depend on the chosen model order.

We have added the following sections to the manuscript:

Results, page 6, lines 159-162:

“This finding does not heavily depend on the chosen model order, given that repeating this analysis using model order 3 (that is a cubic model with 9 TSM coefficients for both the putamen and caudate-NAcc regions – 4x9 TSM coefficients), resulted in a similar correlation (r=0.90, p<0.0001).”

Methods, page 19, lines 587-589:

“In addition, to show that our results do not heavily depend on the chosen model order, we repeated this analysis using model order 3 (that is a cubic model with 9 TSM coefficients) for both the putamen and caudate-NAcc regions (i.e., 4x9=36 TSM coefficients).”

A citation is needed for this assertion: "as PD is known to affect the putamen region of the striatum before the caudate-NAcc region" (l. 215)

We apologize for the missing reference and have now added *Kish et al., 1988*, which is of the earliest studies supporting this claim.